# IRWE: Inductive Random Walk for Joint Inference of Identity and Position Network Embedding

**Meng Qin** *mengqin_az@foxmail.com*
*Department of Computer Science & Engineering, Hong Kong University of Science & Technology*

**Dit-Yan Yeung** *dyyeung@cse.ust.hk*
*Department of Computer Science & Engineering, Hong Kong University of Science & Technology*

**Reviewed on OpenReview:** *https://openreview.net/forum?id=bDse8Z2gff*

## Abstract

Network embedding, which maps graphs to distributed representations, is a unified framework for various graph inference tasks. According to the topology properties (e.g., structural roles and community memberships of nodes) to be preserved, it can be categorized into the identity and position embedding. Most existing methods can only capture one type of property. Some approaches can support the inductive inference that generalizes the embedding model to new nodes or graphs but relies on the availability of attributes. Due to the complicated correlations between topology and attributes, it is unclear for some inductive methods which type of property they can capture. In this study, we explore a unified framework for the joint inductive inference of identity and position embeddings without attributes. An inductive random walk embedding (IRWE) method is proposed, which combines multiple attention units to handle the random walk (RW) on graph topology and simultaneously derives identity and position embeddings that are jointly optimized. We demonstrate that some RW statistics can characterize node identities and positions while supporting the inductive inference. Experiments validate the superior performance of IRWE over various baselines for the transductive and inductive inference of identity and position embeddings.

## 1 Introduction

For various graph inference techniques, network embedding (a.k.a. graph representation learning) is a commonly used framework. It maps each node of a graph to a low-dimensional vector representation (a.k.a. embedding) with some key properties preserved. The derived representations are used to support several downstream inference tasks, e.g., node classification (Kipf & Welling, 2017; Veličković et al., 2018), node clustering (Ye et al., 2022; Qin et al., 2023a; Gao et al., 2023), and link prediction (Lei et al., 2018; 2019; Qin et al., 2023b; Qin & Yeung, 2023).

According to the topology properties to be preserved, existing network embedding techniques can be categorized into the identity and position embedding (Zhu et al., 2021). The identity embedding (a.k.a. structural embedding) preserves the structural role of each node characterized by its rooted subgraph, which is also defined as node identity. The position embedding (a.k.a. proximity-preserving embedding) captures the linkage similarity between nodes in terms of the overlap of local neighbors (i.e., community structures (Newman, 2006)), which is also defined as node position or proximity. In Fig. 1 (a), each color denotes a structural role. For instance, red and yellow may indicate the *opinion leader* and *hole spanner* in a social network (Yang et al., 2015). Moreover, there are two communities denoted by the two dotted circles in Fig. 1, where nodes in the same community have dense linkages and thus are more likely to have similar positions.

The identity and position embedding should respectively force nodes with similar identities (e.g., $\{v_1, v_8\}$) and positions (e.g., $\{v_1, v_2, v_6\}$) to have close embeddings. As a demonstration, we applied *struc2vec* (Ribeiro et al., 2017) and *node2vec* (Grover & Leskovec, 2016) (with embedding dimensionality $d = 2$), which are

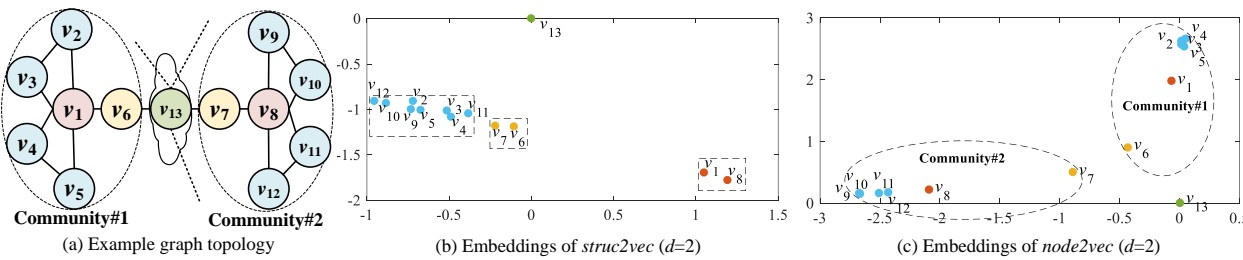

Figure 1: An example of identity and position embedding in terms of (b) *struc2vec* and (c) *node2vec*, where each color denotes a unique identity while nodes in the same community have similar positions.

typical identity and position embedding methods, to the example in Fig. 1 (a) and visualize the derived embeddings. Note that two nodes may have the same identity even though they are far away from each other. In contrast, nodes with similar positions must be close to each other with dense linkage and short distances. Due to the contradiction, *it is challenging to simultaneously capture the two types of properties in a common embedding space.* For instance, $v_1$ and $v_8$ with the same identity have close identity embeddings in Fig. 1 (b). However, their position embeddings are far away from each other in Fig. 1 (c). Since the two types of embeddings may be appropriate for different downstream tasks (e.g., structural role classification and community detection), we expect a unified embedding model.

Most conventional embedding methods (Wu et al., 2020; Grover & Leskovec, 2016; Ribeiro et al., 2017; Donnat et al., 2018) follow the embedding lookup scheme and can only support transductive embedding inference. In this scheme, node embeddings are model parameters optimized only for the currently observed graph topology. When applying the model to new unseen nodes or graphs, one needs to re-train the model from scratch. Compared with transductive methods, some state-of-the-art techniques (Hamilton et al., 2017; Velickovic et al., 2019) can support the advanced inductive inference, which directly generalizes the embedding model trained on observed topology to new unseen nodes or graphs without re-training.

Most existing inductive approaches (e.g., those based on graph neural networks (GNNs) (Wu et al., 2020)) rely on the availability of node attributes and an attribute aggregation mechanism. However, prior studies (Qin et al., 2018; Li et al., 2019; Wang et al., 2020; Qin & Lei, 2021) have demonstrated some complicated correlations between graph topology and attributes. For instance, attributes may provide (i) complementary characteristics orthogonal to topology for better quality of downstream tasks or (ii) inconsistent noise causing unexpected quality degradation. It is unclear for most inductive methods that their performance improvement is brought about by the incorporation of attributes or better exploration of topology. When attributes are unavailable, most inductive approaches require additional procedures to extract auxiliary attribute inputs from topology (e.g., one-hot node degree encodings). Our experiments demonstrate that some inductive baselines with these naive attribute extraction strategies may even fail to outperform conventional transductive methods on the inference of identity and position embeddings. It is also hard to determine which type of properties (i.e., node identities or positions) that some inductive approaches can capture.

In this study, we consider the unsupervised network embedding and explore a unified framework for the joint inductive inference of identity and position embeddings. To clearly distinguish between the two types of embeddings, we consider the case where topology is the only available information source. This eliminates the unclear influence from graph attributes due to the complicated correlations between the two sources. Different from most existing inductive approaches relying on the availability of node attributes, we propose an inductive random walk embedding (IRWE) method. It combines multiple attention units with different choices of key, query, and value to handle the random walk (RW) and induced statistics on graph topology.

RW is an effective technique to explore topology properties for network embedding. However, most RW-based methods (Grover & Leskovec, 2016; Ribeiro et al., 2017) follow the transductive embedding lookup scheme, failing to support the advanced inductive inference. We demonstrate that anonymous walk (AW) (Ivanov & Burnaev, 2018), the anonymization of RW, and its induced statistics can be informative features shared by all possible nodes and graphs and thus have the potential to support inductive inference.

Although the identity and position embedding encodes properties that may contradict with one another, there remains a relation that *nodes with different identities should have different contributions in forming the local community structures*. For the example in Fig. 1, $v_1$ and $v_2$ may correspond to an opinion leader and ordinary audience of a social network, where $v_1$ is expected to contribute more in forming community#1 than $v_2$. By incorporating this relation, IRWE jointly derives and optimizes two sets of embeddings w.r.t. node identities and positions. In particular, we demonstrate that some AW statistics can characterize node identities to derive identity embeddings, which can be further used to generate position embeddings. It is also expected that the joint learning of the two sets of embeddings can improve the quality of one another.

Our major contributions are summarized as follows. (i) In contrast to most existing inductive embedding methods relying on the availability of node attributes, we propose an alternative IRWE approach, whose inductiveness is only supported by the RW on graph topology. (ii) To the best of our knowledge, we are the first to explore a unified framework for the joint inductive inference of identity and position embeddings using RW, AW, and induced statistics. (iii) Experiments on public datasets validate the superiority of IRWE over various baselines for the transductive and inductive inference of identity and position embeddings.

## 2 Related Work

### 2.1 Identity & Position Embedding

In the past several years, a series of network embedding techniques have been proposed. Rossi et al. (2020) gave an overview of existing methods covering the identity and position embedding. Most existing embedding approaches can only capture one type of topology properties (i.e., node identities or positions).

Perozzi et al. (2014) proposed *DeepWalk* that applies skip-gram to learn node embeddings from RWs on graph topology. The ability of *DeepWalk* to capture node positions is further validated in (Pei et al., 2020; Rossi et al., 2020). Grover & Leskovec (2016) modified the RW in *DeepWalk* to a biased form and introduced *node2vec* that can derive richer position embeddings by adjusting the trade-off between breadth- and depth-first sampling. Cao et al. (2015) reformulated the RW in *DeepWalk* to matrix factorization objectives. Wang et al. (2017), Ye et al. (2022), and Chen et al. (2023) introduced community-preserving embedding methods based on nonnegative matrix factorization, hyperbolic embedding, and graph contrastive learning.

Ribeiro et al. (2017) proposed *struc2vec*, an identity embedding method, by applying RW to a multi-layer graph constructed via hierarchical similarities w.r.t. node degrees. Donnat et al. (2018) used graph wavelets to develop *GraphWave* and proved its ability to capture node identities. Pei et al. (2020) introduced *struc2gauss*, which encodes node identities in a space formulated by Gaussian distributions, and analyzed the effectiveness of different energy functions and similarity measures. Guo et al. (2020) enhanced the ability of GNNs to preserve node identities by reconstructing several manually-designed statistics. Chen et al. (2022) enabled the graph transformer to capture node identities by incorporating the rooted subgraph of each node.

Hoff (2007) demonstrated that the latent class and distance models can respectively capture node positions and identities but real networks may exhibit combinations of both properties. An eigen-model was proposed, which can generalize either the latent class model or distance model. However, the proposed eigen-model is a conventional probabilistic model and cannot simultaneously capture both properties in a unified framework. Zhu et al. (2021) proposed a *PhUSION* framework with three steps and showed which components can be used for the identity or position embedding. Although *PhUSION* reveals the similarity and difference between the two types of embeddings, it can only derive one type of embedding under each unique setting. Rossi et al. (2020) validated that some techniques (e.g., RW and attribute aggregation) of existing methods can only derive either identity or position embeddings. Srinivasan & Ribeiro (2020) proved that the relation between identity and position embeddings can be analogous to that of a probability distribution and its samples. Similarly, *PaCEr* (Yan et al., 2024) is a concurrent transductive method that considers the relation between the two types of embeddings based on RW with restart. Although these methods (Srinivasan & Ribeiro, 2020; Yan et al., 2024) can derive both identity and position embeddings, they only involve *the optimization of one type of embedding* and *a simple transform to another type*. In contrast, we focus on the *joint learning and inductive inference of the two types of embeddings*.

### 2.2 Inductive Network Embedding

Some recent studies explore the inductive inference that directly derives embeddings for new unseen nodes or graphs by generalizing the model parameters optimized on known topology. Hamilton et al. (2017) introduced *GraphSAGE*, an inductive GNN framework, including the neighbor sampling and feature aggregation with different choices of aggregation functions. *GAT* (Veličković et al., 2018) leverages self-attention into the attribute aggregation of GNN, which automatically determines the aggregation weights for the neighbors of each node. Velickovic et al. (2019) proposed *DGI* that maximizes the mutual information between patch embeddings and high-level graph summaries. Without using the feature aggregation of GNN, Nguyen et al. (2021) developed *SANNE* that applies self-attention to handle RWs sampled from graph topology. However, the inductiveness of the these methods relies on the availability of node attributes.

Some recent research analyzed the ability of several new GNN structures to capture node identities or positions in specific cases about node attributes (e.g., all the nodes have the same scalar attribute input (Xu et al., 2019)). Wu et al. (2019) and You et al. (2021) proposed *DEMO-Net* and *ID-GNN* that can capture node identities using the degree-specific multi-task graph convolution and heterogeneous message passing on the rooted subgraph of each node, respectively. Jin et al. (2020) leveraged AW statistics into the feature aggregation to enhance the ability of GNN to preserve node identities. *P-GNN* (You et al., 2019) can derive position-aware embeddings based on a distance-weighted aggregation scheme over the sets of sampled anchor nodes. However, these GNN structures can only capture either node identities or positions.

In contrast to the aforementioned methods, we explore *a unified inductive framework for the joint inference of identity and position embeddings without relying on the availability and aggregation of attributes.*

## 3 Problem Statements & Preliminaries

We consider the unsupervised network embedding on undirected unweighted graphs. A graph can be represented as $\mathcal{G} = (\mathcal{V}, \mathcal{E})$, with $\mathcal{V} = \{v_1, v_2, \dots, v_N\}$ and $\mathcal{E} = \{(v_i, v_j) | v_i, v_j \in \mathcal{V}\}$ as the sets of nodes and edges. We also assume that graph topology is the only available information source and attributes are unavailable.

**Definition 1** (*Node Identity*). Node *identity* describes the *structural role* that a node $v$ plays in graph topology (e.g., opinion leader and hole spanner w.r.t. red and yellow nodes in Fig. 1 (a)), which can be characterized by its $l$-hop rooted subgraph $\mathcal{G}_s(v, l)$. Given a pre-set $l$, nodes $(v, u)$ with similar subgraphs $(\mathcal{G}_s(v, l), \mathcal{G}_s(u, l))$ (e.g., measured by the WL graph isomorphism test) are expected to play similar *structural roles* and have similar *identities*.

**Definition 2** (*Node Position*). *Positions* of nodes in graph topology can be encoded by their relative distances and can be further characterized by the linkage similarity in terms of the overlap of $l$-hop neighbors (i.e., community structures). Nodes with a high overlap of $l$-hop neighbors are more likely to (i) have short distance, (ii) belong to the same community, and thus (iii) have similar positions.

**Definition 3** (*Network Embedding*). Given a graph $\mathcal{G}$, we consider the network embedding (a.k.a. graph representation learning) $f : \mathcal{V} \mapsto \mathbb{R}^d$ that maps each node $v$ to a vector $f(v)$ (a.k.a. embedding), with either *node identities* or *positions* preserved. We define $f(v) := \psi(v)$ (or $f(v) := \gamma(v)$) as the *identity* (or *position*) *embedding* if $\{\psi(v)\}$ (or $\{\gamma(v)\}$) preserve node identities (or positions). Namely, nodes $(v, u)$ with similar identities (or positions) should have close representations $(\psi(v), \psi(u))$ (or $(\gamma(v), \gamma(u))$). The learned embeddings are adopted as the inputs of some downstream modules to support concrete inference tasks.

The embedding inference includes the transductive and inductive settings. A transductive method focuses on the optimization of $f$ on the currently observed topology $\mathcal{G} = (\mathcal{V}, \mathcal{E})$ and can only support inference tasks on $\mathcal{V}$. In contrast, an inductive approach can directly generalize its model parameters, which are first optimized on $(\mathcal{V}, \mathcal{E})$, to new unseen nodes $\mathcal{V}'$ or even a new graph $\mathcal{G}'' = (\mathcal{V}'', \mathcal{E}'')$ and support tasks on $\mathcal{V}'$ or $\mathcal{V}''$ (i.e., the inductive inference for new nodes or across graphs). A transductive method cannot support the inductive inference but an inductive approach can tackle both settings.

We focus on the joint inductive inference of identity and position embeddings. A novel IRWE method is proposed which combines multiple attention units to handle RWs and induced AWs.

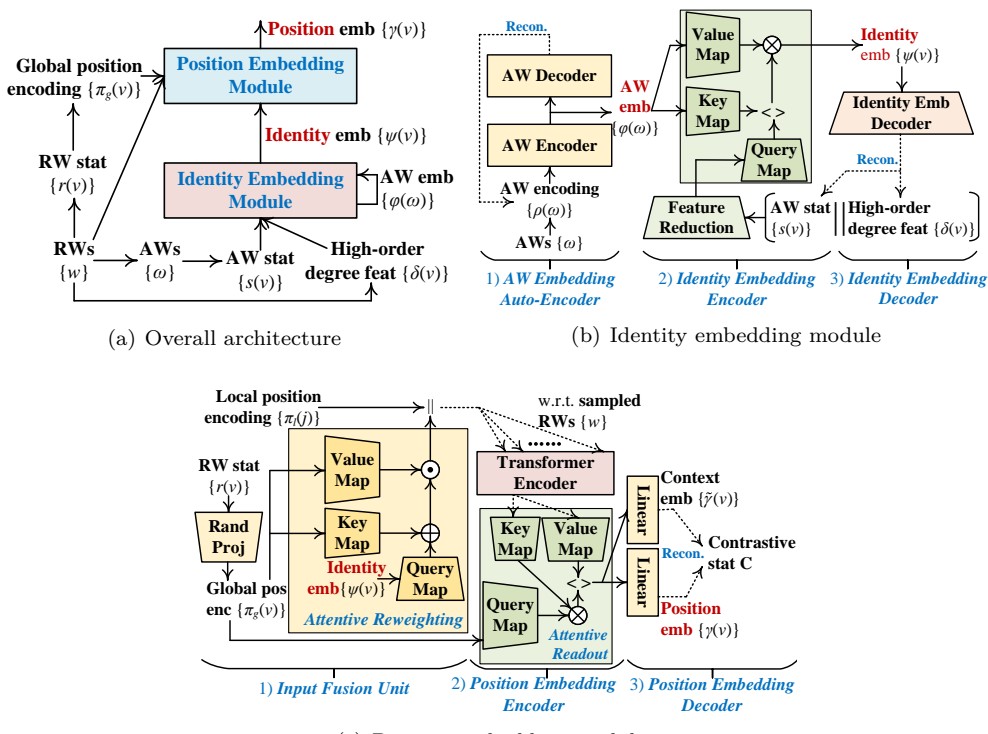

(a) Overall architecture  (b) Identity embedding module

(c) Position embedding module

Figure 2: Model architecture of IRWE including modules of (b) identity and (c) position embeddings.

**Definition 4** (*Random Walk & Anonymous Walk*). An RW with length $l$ is a node sequence $w = (w^{(0)}, w^{(1)}, \ldots, w^{(l)})$, where $w^{(j)} \in \mathcal{V}$ is the $j$-th node and $(w^{(j)}, w^{(j+1)}) \in \mathcal{E}$. Assume that the index $j$ starts from 0. For an RW $w$, one can map it to an AW $\omega = (I_w(w^{(0)}), \ldots, I_w(w^{(l)}))$, where $I_w(w^{(j)})$ maps $w^{(j)}$ to its first occurrence index in $w$.

In Fig. 1 (a), $(v_1, v_4, v_5, v_1, v_6)$ is a valid RW with $(0, 1, 2, 0, 3)$ as its AW. In particular, two RWs (e.g., $(v_1, v_4, v_5, v_1)$ and $(v_8, v_{10}, v_9, v_8)$) can be mapped to a common AW (i.e., $(0, 1, 2, 0)$). In Section 4, we further demonstrate that AW and its induced statistics can be features shared by all possible topology and thus can support the inductive embedding inference without attributes.

## 4 Methodology

In this section, we elaborate on the model architecture as well as the optimization and inference of IRWE. Fig. 2 (a) gives an overview of the model architecture, including two jointly optimized modules that derive identity embeddings $\{\psi(v)\}$ and position embeddings $\{\gamma(v)\}$.

### 4.1 Identity Embedding Module

Fig. 2 (b) highlights details of the identity embedding module. It derives identity embeddings $\{\psi(v)\}$ based on auxiliary AW embeddings $\{\varphi(\omega)\}$, AW statistics $\{s(v)\}$, and high-order degree features $\{\delta(v)\}$. Fig. 3 gives running examples about the extraction of $\{\varphi(\omega)\}$, $\{s(v)\}$, and $\{\delta(v)\}$ based on the local topology of node $v_1$ in Fig. 1, where we set RW length $l = 3$ and the number of sampled RWs $n_S = 5$ as a demonstration. The optimization and inference of this module includes the (1) *AW embedding auto-encoder*, (2) *identity embedding encoder*, and (3) *identity embedding decoder*.

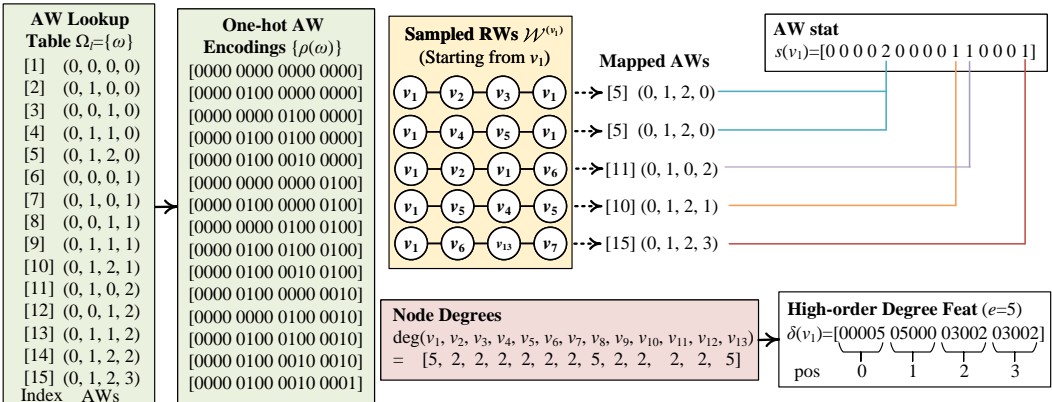

Figure 3: Running examples about the derivation of one-hot AW encodings $\{\rho(\omega)\}$, AW statistics $\{s(v)\}$, and high-order degree features $\{\delta(v)\}$ based on the local topology of node $v_1$ in Fig. 1.

### 4.1.1 AW Embedding Auto-Encoder

As discussed in Section 3, it is possible to map RWs with different sets of nodes to a common AW. For instance, $(0, 1, 2, 0)$ is the common AW of RWs $(v_1, v_2, v_3, v_1)$ and $(v_1, v_4, v_5, v_1)$ in Fig. 3. Given a fixed length $l$, RWs on all possible topology structures can only be mapped to a finite set of AWs $\Omega_l$. Namely, $\Omega_l$ *and its induced statistics are shared by all possible nodes and graphs*, thus having the potential to support the inductive embedding inference. Based on this intuition, IRWE maintains an AW embedding $\varphi(\omega) \in \mathbb{R}^d$ for each AW $\omega \in \Omega_l$. In this setting, $\{\varphi(\omega)\}$ can be used as a special embedding lookup table for the derivation of inductive features regarding graph topology.

We also consider an additional constraint on $\{\varphi(\omega)\}$, where two AWs with more common elements in corresponding positions should have closer representations. For instance, $(0, 1, 2, 1, 2)$ and $(0, 1, 0, 1, 2)$ should be closer in the AW embedding space than $(0, 1, 2, 1, 2)$ and $(0, 1, 0, 2, 3)$. To apply this constraint, we transform each AW $\omega$ with length $l$ to a one-hot encoding $\rho(\omega) \in \{0, 1\}^{(l+1)^2}$, where $\rho(\omega)_{jl:(j+1)l}$ (i.e., subsequence from the $jl$-th to the $(j+1)l$-th positions) is the one-hot encoding of the $j$-th element in $\omega$. For instance, we have $\rho(\omega) = [0000\ 0100\ 0010\ 0001]$ for $\omega = (0, 1, 2, 3)$ in Fig. 3. An auto-encoder is then introduced to derive and regularize $\{\varphi(\omega)\}$, including an encoder and a decoder. Given an AW $\omega$, the encoder $\text{Enc}_\varphi(\cdot)$ and decoder $\text{Dec}_\varphi(\cdot)$ are defined as

$$\varphi(\omega) = \text{Enc}_\varphi(\omega) := \text{MLP}(\rho(\omega)), \ \hat{\rho}(\omega) = \text{Dec}_\varphi(\omega) := \text{MLP}(\varphi(\omega)), \tag{1}$$

which are both multi-layer perceptrons (MLPs). The encoder takes $\rho(\omega)$ as input and derives AW embedding $\varphi(\omega)$. The decoder reconstructs $\rho(\omega)$ with $\varphi(\omega)$ as input. Since similar AWs have similar one-hot encodings, similar AWs can have close embeddings by minimizing the reconstruction error between $\{\rho(\omega)\}$ and $\{\hat{\rho}(\omega)\}$.

### 4.1.2 Identity Embedding Encoder

IRWE derives identity embeddings $\{\psi(v)\}$ via the combination of AW embeddings $\{\varphi(\omega)\}$ inspired by the following **Theorem 1** (Micali & Zhu, 2016).

**Theorem 1.** *Let $\mathcal{G}_s(v, r)$ be the rooted subgraph induced by nodes with a distance less than $r$ from $v$. Let $q(v, l)$ be the distribution of AWs w.r.t. RWs starting from $v$ with length $l$. One can reconstruct $\mathcal{G}_s(v, r)$ in time $O(n^2)$ with $O(n^2)$ access to $[q(v, 1), \cdots, q(v, l)]$, where $l = O(m)$; $n$ and $m$ are the numbers of nodes and edges in $\mathcal{G}_s(v, r)$.*

For a given length $l$, let $\eta_l$ be the number of AWs. $q(v, l)$ can be represented as an $\eta_l$-dimensional vector, with the $j$-th element as the occurrence probability of the $j$-th AW. Since AWs with length $l$ include sequences of those with length less than $l$ (e.g., $(0, 1, 2, 3)$ provides information about $(0, 1, 2)$), one can derive $q(v, k)$ $(k < l)$ based on $q(v, l)$. Therefore, $q(v, l)$ *can be used to characterize $\mathcal{G}_s(v, r)$ according to* **Theorem 1**.

---

**Algorithm 1:** Derivation of AW Statistics

---

**Input:** target node $v$; RW length $l$; sampled RWs $\mathcal{W}^{(v)}$; AW lookup table $\Omega_l$; number of AWs $\eta_l$
**Output:** AW statistic $s(v)$ w.r.t. $v$

1   $s(v) \leftarrow [0, 0, \cdots, 0]^{\eta_l}$ //Initialize $s(v)$
2   **for each** $w \in \mathcal{W}^{(v)}$ **do**
3      map RW $w$ to its AW $\omega$
4      get the index $j$ of AW $\omega$ in lookup table $\Omega_l$
5      $s(v)_j \leftarrow s(v)_j + 1$ //Update $s(v)$

---

As defined in Section 3, nodes with similar rooted subgraphs are expected to play similar structural roles and thus have similar identities. For instance, in Fig. 1, $\mathcal{G}_s(v_1, 1)$ and $\mathcal{G}_s(v_8, 1)$ have the same topology structure, which is consistent with the same identity they have. Hence, $q(v, l)$ *can characterize the identity of node $v$.*

To estimate $q(v, l)$, we extract AW statistic $s(v)$ for each node $v$ using Algorithm 1. We first sample RWs with length $l$ starting from $v$ via the standard unbiased strategy (Perozzi et al., 2014) (see Algorithm 6 in Appendix A). Let $\mathcal{W}^{(v)}$ be the set of sampled RWs starting from $v$. Each RW $w \in \mathcal{W}^{(v)}$ is then mapped to its AW. Let $\Omega_l$ be an AW lookup table including all the $\eta_l$ AWs with length $l$, which *is fixed and shared by all possible topology.* We define the AW statistic as $s(v) := [c(\omega_1), \cdots, c(\omega_{\eta_l})] \in \mathbb{Z}_+^{\eta_l}$, where $c(\omega_j)$ is the frequency of the $j$-th AW in $\Omega_l$ as illustrated in Fig. 3.

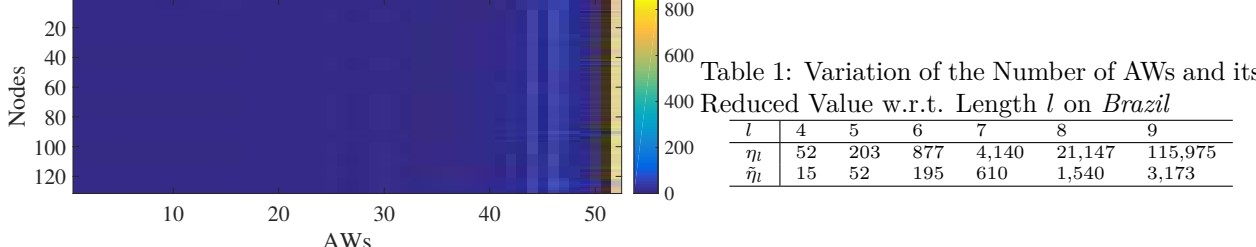

Table 1: Variation of the Number of AWs and its Reduced Value w.r.t. Length $l$ on *Brazil*

| $l$ | 4 | 5 | 6 | 7 | 8 | 9 |
|---|---|---|---|---|---|---|
| $\eta_l$ | 52 | 203 | 877 | 4,140 | 21,147 | 115,975 |
| $\tilde{\eta}_l$ | 15 | 52 | 195 | 610 | 1,540 | 3,173 |

Figure 4: Visualization of AW statistics $\{s(v)\}$ on *Brazil*.

Although $\eta_l$ grows exponentially with the increase of length $l$, $\{s(v)\}$ are usually sparse. Fig. 4 visualizes the example AW statistics $\{s(v)\}$ derived from RWs on the *Brazil* dataset (see Section 5.1 for details) with $l = 4$ and $|\mathcal{W}^{(v)}| = 1,000$. The $i$-th row in Fig. 4 is the AW statistic $s(v_i)$ of node $v_i$. Dark blue indicates that the corresponding element is 0. There exist many AWs $\{\omega_j\}$ not observed during the RW sampling (i.e., $\forall v \in \mathcal{V}$ s.t. $s(v)_j = 0$). We then remove terms w.r.t. these unobserved AWs in $\Omega_l$ and $\{s(v)\}$. Let $\tilde{\Omega}_l$, $\tilde{s}(v)$, and $\tilde{\eta}_l$ be the reduced $\Omega_l$, $s(v)$, and $\eta_l$. Table 1 shows the variation of $\eta_l$ and $\tilde{\eta}_l$ on *Brazil* as $l$ increases from 4 to 9, where $\eta_l$ is significantly reduced.

In addition to $\{\tilde{s}(v)\}$, one can also characterize node identities from the view of node degrees (Ribeiro et al., 2017; Wu et al., 2019) based on the following **Hypothesis 1**.

**Hypothesis 1.** *Nodes with the same degree are expected to play the same structural role. This concept can be extended to high-order neighbors of nodes. Namely, nodes are expected to have similar identities if they have similar node degree statistics (e.g., distribution over all degree values) w.r.t. their high-order neighbors.*

Based on this motivation, we extract high-order degree feature $\delta(v)$ for each node $v$ using Algorithm 2. Given a node $u$, one can construct a bucket one-hot encoding $\rho_d(u) \in \{0, 1\}^e$ w.r.t. its degree $\deg(u)$, where only the $j$-th element $\rho_d(u)_j$ is set to 1 with the remaining elements set to 0 and $j = \lfloor (\deg(u) - \deg_{\min})e / (\deg_{\max} - \deg_{\min}) \rfloor$; $\deg_{\min}$ and $\deg_{\max}$ are the minimum and maximum degrees. Since high-order neighbors of a node $v$ can be explored by RWs $\mathcal{W}^{(v)}$ starting from $v$, we define $\delta(v) \in \mathbb{Z}^{(l+1)e}$ as an $(l+1)e$-dimensional vector, where the subsequence $\delta(v)_{ie:(i+1)e}$ is the sum of bucket one-hot degree encodings w.r.t. nodes occurred at the $i$-th position of RWs in $\mathcal{W}^{(v)}$. Fig. 3 gives a running example to derive $\delta(v_1)$ (with $e = 5$) for node $v_1$ in Fig. 1.

---

**Algorithm 2:** Derivation of Degree Features

**Input:** target node $v$; RW length $l$; one-hot degree encoding dimensionality $e$; sampled RWs $\mathcal{W}^{(v)}$; minimum degree $\deg_{\min}$; maximum degree $\deg_{\max}$
**Output:** high-order degree feature $\delta(v)$ w.r.t. $v$

1  $\delta(v) \leftarrow [0, 0, \cdots, 0]^{(l+1)e}$ //Initialize degree feature $\delta(v)$
2  **for each** $w \in \mathcal{W}^{(v)}$ **do**
3      **for** $i$ **from** $0$ **to** $l$ **do**
4          $u \leftarrow w^{(i)}$ //$i$-th node in current RW $w$
5          $\rho_d(u) \leftarrow [0, \cdots, 0] \in \mathbb{R}^e$ //Initialize degree encoding $\rho_d(u)$
6          $j \leftarrow \lfloor (\deg(u) - \deg_{\min})e/(\deg_{\max} - \deg_{\min}) \rfloor$
7          $\rho_d(u)_j \leftarrow 1$ //Update $\rho_d(u)$
8          $\delta(v)_{ie:(i+1)e} \leftarrow \delta(v)_{ie:(i+1)e} + \rho_d(u)$ //Update $\delta(v)$

---

Following the aforementioned discussions regarding **Theorem 1** and **Hypothesis 1**, *IRWE derives identity embeddings $\{\psi(v)\}$ via the adaptive combination of AW embeddings $\{\varphi(\omega)\}$ w.r.t. AW statistics $\{\tilde{s}(v)\}$ and degree features $\{\delta(v)\}$.* The multi-head attention is applied to automatically determine the contribution of each AW embedding $\varphi(\omega)$ in the combination, where we treat $\{\varphi(\omega)\}$ as the key and value; the concatenated feature $[\tilde{s}(v)||\delta(v)]$ is used as the query. Before feeding $[\tilde{s}(v)||\delta(v)] \in \mathbb{R}^{(\bar{\eta}_l + le)}$ to the multi-head attention, we introduce a feature reduction unit $\mathrm{Red}_s(\cdot)$, an MLP, to reduce its dimensionality to $d$:

$$\bar{g}(v) = \mathrm{Red}_s(v) := \mathrm{MLP}([\tilde{s}(v)||\delta(v)]). \tag{2}$$

The multi-head attention that derives identity embeddings $\{\psi(v)\}$ is defined as

$$\mathbf{Z} = \mathrm{Att}(\mathbf{Q}, \mathbf{K}, \mathbf{V}) = \mathrm{Att}(\{\bar{g}(v)\}, \{\varphi(\omega)\}, \{\varphi(\omega)\}), \tag{3}$$

where $\mathrm{Att}(\cdot, \cdot, \cdot)$ is the standard multi-head attention unit (see Appendix D for details), with $\mathbf{Q}$, $\mathbf{K}$, and $\mathbf{V}$ as inputs of query, key, and value. In (3), we have $\mathbf{Q}_{i,:} = \bar{g}(v_i)$, $\mathbf{K}_{j,:} = \mathbf{V}_{j,:} = \varphi(\omega_j)$, and $\mathbf{Z}_{i,:} = \psi(v_i)$.

### 4.1.3 Identity Embedding Decoder

An identity embedding decoder $\mathrm{Dec}_\psi(\cdot)$ is introduce to regularize identity embeddings $\{\psi(v)\}$ using statistics $\{[\tilde{s}(v)||\delta(v)]\}$. It takes the $\psi(v)$ of a node $v$ as input and reconstruct corresponding $[\tilde{s}(v)||\delta(v)]$ via

$$\hat{g}(v) = \mathrm{Dec}_\psi(v) := \mathrm{MLP}(\psi(v)), \tag{4}$$

where $\hat{g}(v)$ is the reconstructed statistic. It can force $\{\psi(v)\}$ to capture node identities hidden in $\{[\tilde{s}(v)||\delta(v)]\}$ by minimizing the reconstruction error between $\{\hat{g}(v)\}$ and $\{[\tilde{s}(v)||\delta(v)]\}$. Note that we only apply $\mathrm{Dec}_\psi(\cdot)$ to optimize $\{\psi(v)\}$ and do not need this unit in the inference phase.

## 4.2 Position Embedding Module

Fig. 2 (c) gives an overview of the position embedding module. It derives position embeddings $\{\gamma(v)\}$ based on (i) identity embeddings $\{\psi(v)\}$ given by the previous module and (ii) auxiliary position encodings $\{\pi_g(v), \pi_l(j)\}$ extracted from the sampled RWs $\{\mathcal{W}^{(v)}\}$. Instead of using the attribute aggregation mechanism of GNNs, we convert the graph topology into a set of RWs and use the transformer encoder (Vaswani et al., 2017), a sophisticated structure that can process sequential data, to handle RWs. Besides the sequential input, transformer also requires a 'position' encoding to describe the position of each element in the sequence. As highlighted in **Definition 3**, the physical meaning of node position in non-Euclidean graphs is different from that in Euclidean sequences (e.g., sentences and RWs). To describe the (i) Euclidean position in RWs and (ii) node position in a graph, we introduce the *local* and *global* position encodings (denoted as $\pi_l(j)$ and $\pi_g(v)$) for a sequence position with index $j$ and each node $v$. The optimization and inference of this module includes the (1) *input fusion unit*, (2) *position embedding encoder*, and (3) *position embedding decoder*.

---

**Algorithm 3:** Derivation of Global Position Encoding

---

**Input:** target node $v$; sampled RWs $\mathcal{W}^{(v)}$; node set $\mathcal{V}$; random matrix $\boldsymbol{\Theta} \in \mathbb{R}^{|\mathcal{V}| \times d}$

**Output:** global position encoding $\pi_g(v)$ w.r.t. $v$

1   $r(v) \leftarrow [0, 0, \cdots, 0]^{|\mathcal{V}|}$ //Initialize RW statistic $r(v)$

2   **for each** $w \in \mathcal{W}^{(v)}$ **do**

3     **for each** $v_j \in w$ **do**

4       $r(v)_j \leftarrow r(v)_j + 1$ //Update $r(v)$

5   $\pi_g(v) \leftarrow r(v)\boldsymbol{\Theta}$ //Derive $\pi_g(v)$

---

#### 4.2.1 Input Fusion Unit

The *input fusion unit* extracts $\{\pi_l(j), \pi_g(v)\}$ and derives inputs of the transformer encoder combined with $\{\psi(v)\}$. Since the RW length $l$ is usually not very large (e.g., $\leq 10$ in our experiments), we define the local position encoding $\pi_l(j) \in \{0, 1\}^{l+1}$ as the standard one-hot encoding of index $j$. Inspired by previous studies (Perozzi et al., 2014; Grover & Leskovec, 2016; Zhu et al., 2021) that validated the potential of RW for exploring local community structures, we extract the global position encoding $\{\pi_g(v)\}$ for each node $v$ w.r.t. RW statistic $r(v)$ using Algorithm 3.

Given a node $v$, we maintain $r(v) \in \mathbb{Z}^{|\mathcal{V}|}$, with the $j$-th element $r(v)_j$ as the frequency that node $v_j$ occurs in RWs $\mathcal{W}^{(v)}$ starting from $v$. For instance, we have $r(v_1) = [8, 2, 1, 2, 3, 2, 1, 0, 0, 0, 0, 0, 1]$ for the running example in Fig. 3 with 13 nodes. Since nodes in a community are densely connected, *nodes within the same community are more likely to be reached via RWs*. Therefore, *nodes $(v, u)$ with similar positions are expected to have similar statistics $(r(v), r(u))$*. We then derive $\pi_g(v)$ by mapping $r(v)$ to a $d$-dimensional vector via the following Gaussian random projection, an efficient dimension reduction technique that can preserve the relative distance between input features with a rigorous guarantee (Arriaga & Vempala, 2006):

$$\pi_g(v) = r(v)\boldsymbol{\Theta} \text{ with } \boldsymbol{\Theta} \in \mathbb{R}^{|\mathcal{V}| \times d}, \boldsymbol{\Theta}_{ir} \sim \mathcal{N}(0, 1/d). \tag{5}$$

In this setting, the non-Euclidean node positions in a graph topology are encoded in terms of the relative distance between $\{\pi_g(v)\}$. Hence, $\pi_g(v)$ has the initial ability to encode the position of node $v$. IRWE integrates the relation between node identities and positions based on the following **Hypothesis 2**.

**Hypothesis 2**. *In a community (i.e., node cluster with dense linkages), nodes with different structural roles may have different contributions in forming the community structure.*

For instance, in a social network, an opinion leader (e.g., $v_1$ and $v_8$ in Fig. 1) is expected to have more contributions in forming the community it belongs to than an ordinary audience (e.g., $v_2$ and $v_9$ in Fig. 1). Based on this intuition, we use identity embeddings $\{\psi(v)\}$ to reweight global position encodings $\{\pi_g(v)\}$, with the reweighting contributions determined by a modified attention operation. Concretely, we set identity embeddings $\{\psi(v)\}$ as the query and let global position encodings $\{\pi_g(v)\}$ as the key and value (i.e., $\mathbf{Q}_{i,:} = \psi(v_i)$ and $\mathbf{K}_{i,:} = \mathbf{V}_{i,:} = \pi_g(v_i)$). The modified attention operation is defined as

$$\mathbf{Z} = \text{ReAtt}(\mathbf{Q}, \mathbf{K}, \mathbf{V}) := (\text{MLP}(\tilde{\mathbf{Q}}) + \text{MLP}(\tilde{\mathbf{K}})) \odot \tilde{\mathbf{V}}, \tag{6}$$

where $\tilde{\mathbf{Q}} := \text{BN}(\mathbf{Q})$, $\tilde{\mathbf{K}} := \text{BN}(\mathbf{K})$, and $\tilde{\mathbf{V}} := \text{BN}(\mathbf{V})$; $\text{BN}(\cdot)$ and $\odot$ are the batch normalization and element-wise multiplication. In (6), we apply two MLPs to derive nonlinear mappings of the normalized $\{\mathbf{Q}, \mathbf{K}\}$ and use their sum to support the *element-wise reweighting* of the normalized $\mathbf{V}$. For convenience, we denote the reweighted vector w.r.t. a node $v_i$ as $\bar{\pi}_g(v_i) = \mathbf{Z}_{i,:}$. Given an RW $w = (w^{(0)}, w^{(1)}, \cdots, w^{(l)})$, IRWE concatenates the reweighted vector $\bar{\pi}_g(w^{(j)})$ and local position encoding $\pi_l(j)$ for the $j$-th node and feeds its linear mapping to the transformer encoder:

$$t(w^{(j)}) := [\bar{\pi}_g(w^{(j)}) || \pi_l(j)] \mathbf{W}_t, \tag{7}$$

where $\mathbf{W}_t \in \mathbb{R}^{(d+l+1) \times d}$ is a trainable parameter.

### 4.2.2 Position Embedding Encoder

IRWE uses the transformer encoder $\text{TransEnc}(\cdot)$ to handle an RW $w = (w^{(0)}, \cdots, w^{(l)})$:

$$(\bar{t}(w^{(0)}), \cdots) = \text{TransEnc}(t(w^{(0)}), \cdots, t(w^{(l)})). \tag{8}$$

It takes the corresponding sequence of vectors $(t(w^{(0)}), \cdots, t(w^{(l)}))$ as input and derives another sequence of vectors $(\bar{t}(w^{(0)}), \cdots, \bar{t}(w^{(l)}))$ with the same dimensionality. $\text{TransEnc}(\cdot)$ follows a multi-layer structure, with each layer including the self-attention, skip connections, layer normalization, and feedforward mapping. Due to space limit, we omit details of $\text{TransEnc}(\cdot)$ that can be found in (Vaswani et al., 2017).

For an RW $w$ starting from a node $v$, the first output vector $\bar{t}(w^{(0)}) = \bar{t}(v)$ can be a representation of $v$. As we sample multiple RWs $\mathcal{W}^{(v)}$ starting from each node $v$, one can obtain multiple such representations based on $\mathcal{W}^{(v)}$. However, we only need one unique positon embedding $\gamma(v)$ for $v$. Let $\bar{t}^{(v)} := \{\bar{t}(w^{(0)})|w \in \mathcal{W}^{(v)}\}$. A naive strategy to derive $\gamma(v)$ is to average representations in $\bar{t}^{(v)}$. Instead, we develop the following *attentive readout function* to compute the weighted mean of $\bar{t}^{(v)}$, with the weights determined by attention:

$$\mathbf{z} = \text{ROut}(\bar{t}^{(v)}, \pi_g(v)) := \text{Att}(\pi_g(v), \bar{t}^{(v)}, \bar{t}^{(v)}), \ \gamma(v) := \mathbf{z}\mathbf{W}_\gamma + \mathbf{b}_\gamma, \text{ and } \bar{\gamma}(v) := \mathbf{z}\mathbf{W}_{\bar{\gamma}} + \mathbf{b}_{\bar{\gamma}}. \tag{9}$$

In (9), $\text{Att}(\cdot, \cdot, \cdot)$ is the standard multi-head attention unit (see Appendix D for details), where we let the global position encoding $\pi_g(v)$ be the query (i.e., $\mathbf{Q} = \pi_g(v) \in \mathbb{R}^{1 \times d}$) and $\bar{t}^{(v)}$ be the key and value (i.e., $\mathbf{K}_{j,:} = \mathbf{V}_{j,:} = \bar{t}_j^{(v)}$). $\gamma(v)$ and $\bar{\gamma}(v)$ are the (i) *position embedding* and (ii) auxiliary *context embedding* of node $v$, with $\{\mathbf{W}_\gamma, \mathbf{b}_\gamma, \mathbf{W}_{\bar{\gamma}}, \mathbf{b}_{\bar{\gamma}}\}$ as trainable parameters.

### 4.2.3 Position Embedding Decoder

The *position embedding decoder* is introduced to optimize position embeddings $\{\gamma(v)\}$ together with auxiliary context embeddings $\{\bar{\gamma}(v)\}$. Some of existing embedding methods (Perozzi et al., 2014; Tang et al., 2015; Hamilton et al., 2017) are optimized via the following contrastive loss with negative sampling:

$$\min \mathcal{L}_{\text{cnr}} = -\sum\nolimits_{(v_i,v_j) \in D} [p_{ij} \ln \sigma(\gamma(v_i)\tilde{\gamma}^T(v_j)/\tau) + Qn_j \ln \sigma(-\gamma(v_i)\tilde{\gamma}^T(v_j)/\tau)], \tag{10}$$

where $D$ denotes the training set including positive and negative samples in terms of node pairs $\{(v_i, v_j)\}$; $p_{ij}$ is defined as the statistic of a positive node pair $(v_i, v_j)$ (e.g., the frequency that $(v_i, v_j)$ occurs in the RW sampling); $Q$ is the number of negative samples; $n_j$ is usually set to be the probability that $(v_i, v_j)$ is selected as a negative sample; $\sigma(\cdot)$ is the sigmoid function; $\tau$ is a temperature parameter to be specified. We follow prior work (Tang et al., 2015) to let $p_{ij} := \mathbf{A}_{ij}/\deg(v_i)$ (i.e., the probability that there is an edge from $v_i$ to $v_j$ with $\mathbf{A} \in \{0, 1\}^{|\mathcal{V}| \times |\mathcal{V}|}$ as the adjacency matrix) and $n_j \propto (\sum_{i:(v_i,v_j) \in \mathcal{E}} p_{ij})^{0.75}$. In the next section, we demonstrate that the contrastive loss (10) can be converted to a reconstruction loss such that the joint optimization of IRWE only includes several reconstruction objectives.

### 4.3 Model Optimization & Inference

Given an RW length $l$, let $\tilde{\Omega}_l$ be the reduced AW lookup table w.r.t. the reduced AW statistics $\{\tilde{s}(v)\}$ in (2). The optimization objective of identity embeddings $\{\psi(v)\}$ can be described as

$$\min \mathcal{L}_\psi := \mathcal{L}_{\text{reg}-\varphi} + \alpha\mathcal{L}_{\text{reg}-\psi}, \tag{11}$$

$$\mathcal{L}_{\text{reg}-\varphi} := \sum\nolimits_{\omega \in \tilde{\Omega}_l} |\rho(\omega) - \hat{\rho}(\omega)|_2^2, \tag{12}$$

$$\mathcal{L}_{\text{reg}-\psi} := \sum\nolimits_{v \in \mathcal{V}} |[\tilde{s}(v)||\delta(v)]/|\mathcal{W}^{(v)}| - \hat{g}(v)|_2^2, \tag{13}$$

where $\mathcal{L}_{\text{reg}-\varphi}$ regularizes auxiliary AW embeddings $\{\varphi(\omega)\}$ by reconstructing the one-hot AW encodings $\{\rho(\omega)\}$ via the auto-encoder defined in (1); $\mathcal{L}_{\text{reg}-\psi}$ regularizes the derived identity embeddings $\{\psi(v)\}$ by minimizing the error between (i) features $\{[\tilde{s}(v)||\delta(v)]\}$ normalized by the number of sampled RWs $|\mathcal{W}^{(v)}|$ and (ii) reconstructed values $\{\hat{g}(v)\}$ given by (4); $\alpha$ is a tunable parameter.

---

**Algorithm 4:** Model Optimization of IRWE

---

**Input:** topology $(\mathcal{V}, \mathcal{E})$; RW settings $\{l, n_S, n_I\}$; local position encodings $\{\pi_l(j)\}$; optimization settings $\{m, m_\psi, m_\gamma, \lambda_\psi, \lambda_\gamma\}$

**Output:** sampled RWs $\{\mathcal{W}^{(v)}, \mathcal{W}_I^{(v)}\}$; reduced AW lookup table $\tilde{\Omega}_l$ & induced statistics $\{\tilde{s}(v), \delta(v), \pi_g(v)\}$; optimized model parameters $\{\theta_\psi^*, \theta_\gamma^*\}$

**1** get AW lookup table $\Omega_l$ w.r.t. length $l$

**2** get min degree $\deg_{\min}$ & max degree $\deg_{\max}$ of $(\mathcal{V}, \mathcal{E})$

**3** get contrastive statistic $\mathbf{C}$

**4 for each** *node* $v \in \mathcal{V}$ **do**

**5**     sample $n_S$ RWs $\mathcal{W}^{(v)}$ staring from $v$ via Algorithm 6

**6**     get AW statistic $s(v)$ w.r.t. $\mathcal{W}^{(v)}$ via Algorithm 1

**7**     get degree feature $\delta(v)$ w.r.t. $\{\mathcal{W}^{(v)}, d_{\min}, d_{\max}\}$ via Algorithm 2

**8**     get global position encoding $\pi_g(v)$ w.r.t. $\mathcal{W}^{(v)}$ via Algorithm 3

**9**     randomly select $n_I$ RWs $\mathcal{W}_I^{(v)}$ from $\mathcal{W}^{(v)}$

**10** get reduced AW statistic $\{\tilde{s}(v)\}$ by deleting unobserved AWs

**11** get reduced AW lookup table $\tilde{\Omega}_l$ w.r.t. $\{\tilde{s}(v)\}$

**12** initial model parameters $\{\theta_\psi, \theta_\gamma\}$

**13 for** *iter_count* **from** 1 **to** $m$ **do**

**14**     **for** *count$_\psi$* **from** 1 **to** $m_\psi$ **do**

**15**        get $\{\hat{\rho}(\omega), \hat{g}(v)\}$ w.r.t. $\{\tilde{\Omega}_l, \tilde{s}(v), \delta(v)\}$

**16**        get training loss $\mathcal{L}_\psi$ via (11)

**17**        optimize *identity embeddings* $\{\psi(v)\}$ via $\mathrm{Opt}(\lambda_\psi, \theta_\psi, \mathcal{L}_\psi)$

**18**     **for** *count$_\gamma$* **from** 1 **to** $m_\gamma$ **do**

**19**        get identity embeddings $\{\psi(v)\}$ w.r.t. $\{\tilde{\Omega}_l, \tilde{s}(v), \delta(v)\}$

**20**        get position embeddings $\{\gamma(v)\}$ w.r.t. $\{\psi(v), \pi_g(v), \pi_l(j), \mathcal{W}_I^{(v)}\}$

**21**        get training loss $\mathcal{L}_\gamma$ via (14)

**22**        optimize *position embeddings* $\{\gamma(v)\}$ via $\mathrm{Opt}(\lambda_\gamma, \{\theta_\psi, \theta_\gamma\}, \mathcal{L}_\gamma)$

**23**     save model parameters $\{\theta_\psi, \theta_\gamma\}$

---

As described in Section 4.2.3, one can optimize position embeddings $\{\gamma(v)\}$ via a contrastive loss (10). It can be converted to another reconstruction loss based on the following **Proposition 1**. In this setting, the optimization of $\{\psi(v)\}$ and $\{\gamma(v)\}$ only includes three simple reconstruction losses.

**Proposition 1.** *Let* $\mathbf{\Gamma} \in \mathbb{R}^{|\mathcal{V}| \times d}$ *and* $\bar{\mathbf{\Gamma}} \in \mathbb{R}^{|\mathcal{V}| \times d}$ *be the matrix forms of* $\{\gamma(v_i)\}$ *and* $\{\bar{\gamma}(v_i)\}$ *with the i-th rows denoting the corresponding embeddings of node* $v_i$. *We introduce the auxiliary contrastive statistics* $\mathbf{C} \in \mathbb{R}^{|\mathcal{V}| \times |\mathcal{V}|}$ *in terms of a sparse matrix where* $\mathbf{C}_{ij} = \ln p_{ij} - \ln(Qn_j)$ *if* $(v_i, v_j) \in \mathcal{E}$ *and* $\mathbf{C}_{ij} = 0$ *otherwise.* *The contrastive loss (10) is equivalent to the following reconstruction loss:*

$$\min \mathcal{L}_\gamma = \left\| \mathbf{\Gamma}\bar{\mathbf{\Gamma}}^T/\tau - \mathbf{C} \right\|_F^2. \tag{14}$$

The key idea to prove **Proposition 1** is to let the partial derivative $\partial\mathcal{L}_{\mathrm{cnr}}/\partial[\gamma(v_i)\bar{\gamma}^T(v_j)/\tau]$ w.r.t. each edge $(v_i, v_j)$ to 0. We leave the proof of **Proposition 1** in Appendix C.

Algorithm 4 summarizes the joint optimization procedure of IRWE. Before formally optimizing the model, we sampled $n_S$ RWs $\mathcal{W}^{(v)}$ starting from each node $v$ and derive statistics $\{\tilde{s}(v), \delta(v), \pi_g(v)\}$ induced by $\{\mathcal{W}^{(v)}\}$. In particular, we randomly select $n_I$ RWs $\mathcal{W}_I^{(v)}$ from $\mathcal{W}^{(v)}$ ($n_I < n_S$) for each node, which are handled by the transformer encoder in the position embedding module. Namely, we use a ratio of the sampled RWs to derive $\{\gamma(v)\}$ due to the high complexity of transformer. We only sample RWs and derive induced statistics once, which are shared by the following optimization iterations.

To jointly optimize $\{\psi(v)\}$ and $\{\gamma(v)\}$, one can combine (11) and (14) to derive a single hybrid optimization objective. Our pre-experiments show that better embedding quality can be achieved if we separately optimize the two types of embeddings. One possible reason is that the two modules have unbalanced scales of parameters. Let $\theta_\psi$ and $\theta_\gamma$ be the sets of model parameters of the identity and position modules. The scale

---

**Algorithm 5:** Inductive Inference within a Graph

---

**Input:** optimized model parameters $\{\theta_\psi^*, \theta_\gamma^*\}$; new topology $(\mathcal{V} \cup \mathcal{V}', \mathcal{E}')$; RW settings $\{l, n_S, n_I\}$; local position encodings $\{\pi_l(j)\}$; $\{\tilde{\Omega}_l, \deg_{\min}, \deg_{\max}, \tilde{s}(v), \delta(v), \pi_g(v)\}$ derived in model optimization on old topology $(\mathcal{V}, \mathcal{E})$

**Output:** *inductive* embeddings $\{\psi(v)\}$ & $\{\gamma(v)\}$ w.r.t. $\mathcal{V}'$

**1 for each** *node* $v \in \mathcal{V}'$ **do**

**2**      sample $n_S$ RWs $\mathcal{W}^{(v)}$ from $v$ w.r.t. $\mathcal{E}'$ via Algorithm 6

**3**      get AW statistic $\tilde{s}'(v)$ w.r.t. $\{\mathcal{W}^{(v)}, \tilde{\Omega}_l\}$ via Algorithm 8

**4**      get degree feature $\delta'(v)$ w.r.t. $\{\mathcal{W}^{(v)}, \mathrm{d}_{\min}, \mathrm{d}_{\max}\}$ via Algorithm 9

**5**      get global position encoding $\pi_g'(v)$ w.r.t. $\{\mathcal{W}^{(v)}, \mathcal{V}\}$ via Algorithm 10

**6**      randomly select $n_I$ RWs $\mathcal{W}_I^{(v)}$ from $\mathcal{W}^{(v)}$

**7**      add $\tilde{s}'(v)$, $\delta'(v)$, & $\pi_g'(v)$ to $\{\tilde{s}(v)\}$, $\{\delta(v)\}$, & $\{\pi_g(v)\}$

**8** get $\{\psi(v)\}$ based on $\{\tilde{\Omega}_l, \tilde{s}(v), \delta(v)\}$ w.r.t. $\mathcal{V} \cup \mathcal{V}'$

**9** get $\{\gamma(v)\}$ based on $\{\psi(v), \pi_g(v), \pi_l(j), \mathcal{W}_I^{(v)}\}$ w.r.t. $\mathcal{V}'$

---

of $\theta_\gamma$ is larger than $\theta_\psi$ due to the application of transformer. As described in lines 14-17 and lines 19-22, we respectively update $\{\psi(v)\}$ and $\{\gamma(v)\}$ $m_\psi \geq 1$ and $m_\gamma \geq 1$ times based on (11) and (14) in each iteration, where we can balance the optimization of $\{\psi(v)\}$ and $\{\gamma(v)\}$ by adjusting $m_\psi$ and $m_\gamma$.

Note that $\{\psi(v)\}$ are inputs of the position embedding module, providing node identity information for the inference of $\{\gamma(v)\}$. The optimization of $\{\gamma(v)\}$ also includes the update of $\theta_\psi$ via gradient descent, which also affect the inference of $\{\psi(v)\}$. Therefore, *the two types of embeddings are jointly optimized although we adopt a separate updating strategy.* The Adam optimizer is used to update $\{\theta_\psi, \theta_\gamma\}$, with $\lambda_\psi$ and $\lambda_\gamma$ as the learning rates for $\{\psi(v)\}$ and $\{\gamma(v)\}$. Finally, we save model parameters after $m$ iterations.

During the model optimization, we save the sampled RWs $\{\mathcal{W}^{(v)}, \mathcal{W}_I^{(v)}\}$, reduced AW lookup table $\tilde{\Omega}_l$, and induced statistics $\{\tilde{s}(v), \delta(v), \pi_g(v)\}$ (i.e., lines 4-11 in Algorithm 4) and use them as inputs of the *transductive inference* of $\{\psi(v)\}$ and $\{\gamma(v)\}$. Then, the *transductive inference* only includes one feedforward propagation through the model. We summarize this simple inference procedure in Algorithm 7 (see Appendix A).

To support the *inductive inference for new nodes within a graph*, we adopt an incremental strategy to get the inductive statistics $\{\tilde{s}(v), \delta(v), \pi_g(v)\}$ via modified versions of Algorithms 1, 2, and 3 that utilize some intermediate results derived during the training on old topology $(\mathcal{V}, \mathcal{E})$. Algorithm 5 summarizes the *inductive inference within a graph*. Let $\mathcal{V}'$ and $\mathcal{E}'$ be the set of new nodes and edge set induced by $\mathcal{V} \cup \mathcal{V}'$. We sample RWs $\mathcal{W}^{(v)}$ for each new node $v \in \mathcal{V}'$ and get the AW statistic $\tilde{s}(v)$ w.r.t. AWs in the lookup table $\tilde{\Omega}_l$ reduced on old topology $(\mathcal{V}, \mathcal{E})$ rather than all AWs. $\delta(v)$ is derived based on the one-hot degree encoding truncated by the minimum and maximum degrees of $(\mathcal{V}, \mathcal{E})$. In the derivation of $\pi_g(v)$, we compute truncated RW statistic $r(v)$ only w.r.t. previously observed nodes $\mathcal{V}$. We detail procedures to derive inductive $\{\tilde{s}(v), \delta(v), \pi_g(v)\}$ in Algorithms 8, 9, and 10 (see Appendix A). Similar to the *transductive inference*, given the derived $\{\tilde{s}(v), \delta(v), \pi_g(v)\}$, we obtain the *inductive* $\{\psi(v)\}$ and $\{\gamma(v)\}$ via one feedforward propagation.

For the *inductive inference across graphs*, we sample RWs $\{\mathcal{W}^{(v)}, \mathcal{W}_I^{(v)}\}$ on each new graph $(\mathcal{V}'', \mathcal{E}'')$. Since there are no shared nodes between the training and inference topology, we only incrementally compute the reduced/truncated statistics $\{\tilde{s}(v), \delta(v)\}$ using the procedures of lines 3-4 in Algorithm 5. We derive global position encodings $\{\pi_g(v)\}$ from scratch via Algorithm 3. We summarize this *inductive inference* procedure in Algorithm 11 (see Appendix A). We also leave detailed complexity analysis of IRWE in Appendix B.

## 5 Experiments

In this section, we elaborate on our experiments. Section 5.1 introduces experiment setups. Evaluation results for the transductive and inductive embedding inference are described and analyzed in Sections 5.2 and 5.3. Ablation study and parameter analysis are introduced in Sections 5.4 and 5.5. Due to space limit, we leave detailed experiment settings and further experiment results in Appendix D and E.

## 5.1 Experiment Setups

Table 3: Details of Methods to be Evaluated

| Methods | Trans | Ind | Pos | Ide |
|---|---|---|---|---|
| node2vec (Grover & Leskovec, 2016) | √ | | √ | |
| GraRep (Cao et al., 2015) | √ | | √ | |
| struc2vec (Ribeiro et al., 2017) | √ | | | √ |
| struc2gauss (Pei et al., 2020) | √ | | | √ |
| PaCEr (Yan et al., 2024) | √ | | Δ | Δ |
| PhUSION (Zhu et al., 2021) | √ | | Δ | Δ |
| GraphSAGE (Hamilton et al., 2017) | | √ | - | - |
| DGI (Velickovic et al., 2019) | | √ | - | - |
| GraphMAE (Hou et al., 2022) | | √ | - | - |
| GraphMAE2 (Hou et al., 2023) | | √ | - | - |
| P-GNN (You et al., 2019) | | √ | √ | |
| CSGCL (Chen et al., 2023) | | √ | √ | |
| GraLSP (Jin et al., 2020) | | √ | | √ |
| SPINE (Guo et al., 2019) | | √ | | √ |
| GAS (Guo et al., 2020) | | √ | | √ |
| SANNE (Nguyen et al., 2021) | | √ | - | - |
| UGFormer (Nguyen et al., 2022) | | √ | - | - |
| **IRWE** (ours) | | √ | √ | √ |

Table 2: Statistics of Datasets

| Datasets | N | E | K |
|---|---|---|---|
| PPI | 3,890 | 38,739 | 50 |
| Wiki | 4,777 | 92,517 | 40 |
| BlogCatalog | 10,312 | 333,983 | 39 |
| USA | 1,190 | 13,599 | 4 |
| Europe | 399 | 5,993 | 4 |
| Brazil | 131 | 1,003 | 4 |
| PPIs | 1,021-3,480 | 4,554-26,688 | 10 |

**Datasets**. We used seven datasets commonly used by related research to validate the effectiveness of IRWE, with statistics shown in Table 2, where $N$, $E$, and $K$ are the numbers of nodes, edges, and classes.

*PPI*, *Wiki*, and *BlogCatalog* are the first type of datasets (Grover & Leskovec, 2016; Zhu et al., 2021) providing the ground-truth of node positions for multi-label classification. *USA*, *Europe*, and *Brazil* are the second type of datasets (Ribeiro et al., 2017; Zhu et al., 2021) with node identity ground-truth for multi-class classification. In summary, *PPI*, *Wiki*, and *BlogCatalog* are widely used to evaluate the quality of *position embedding* while *USA*, *Europe*, and *Brazil* are well-known datasets for the evaluation of *identity embedding*.

*PPIs* is a widely used dataset for the *inductive inference across graphs* (Hamilton et al., 2017; Veličković et al., 2018), which includes a set of protein-protein interaction graphs (in terms of connected components). In addition to graph topology, *PPIs* also provides node features and ground-truth for node classification. As stated in Section 3, we do not consider graph attributes due to the complicated correlations between topology and attributes. It is also unclear whether the classification ground-truth is dominated by topology or attributes. Therefore, we only used the graph topology of *PPIs*.

**Downstream Tasks**. We adopted multi-label and multi-class node classification for the evaluation of position and identity embeddings on the first and second types of datasets, respectively. In particular, each node may belong to multiple classes in multi-label classification while each node only belongs to one class in multi-class classification. We used Micro *F1-score* as the quality metric for the two classification tasks. To avoid the exception that some labels are not presented in all training examples, we removed classes with very few numbers of members (i.e., less than 8) when conducting node classification.

We also adopted unsupervised node clustering to evaluate the quality of identity and position embeddings. Inspired by spectral clustering (Von Luxburg, 2007) and **Hypothesis 1**, we constructed an auxiliary (top-10) similarity graph $\mathcal{G}_D$ based on the high-order degree features $\{\delta'(v) \in \mathbb{R}^{(l+1)e}\}$ derived via a procedure similar to Algorithm 2. The only difference between $\{\delta'(v)\}$ (used for evaluation) and $\{\delta(v)\}$ (used in IRWE) is that $\delta'(v)$ is derived from the rooted subgraph $\mathcal{G}_s(v, l')$ but not the sampled RWs $\mathcal{W}^{(v)}$. To obtain $\{\delta'(v)\}$, we set $l = 5$ (i.e., the order of neighbors) and $e = 500$ (i.e., the dimensionality of the one-hot degree encoding) for the first type of datasets while we let $l = 3$ and $e = 200$ for *PPIs*. Namely, we applied a clustering algorithm to embeddings learned on the original graph $\mathcal{G}$ but evaluated the clustering result on $\mathcal{G}_D$. We define this task as the *node identity clustering* and expect that it can measure the quality of identity embeddings becuase high-order degree features $\{\delta'(v)\}$ can capture node identities.

In addition, we treated the node clustering evaluated on the original graph $\mathcal{G}$ as *community detection* (Newman, 2006), a task commonly used for the evaluation of position embeddings. *Normalized cut* (*NCut*) (Von Luxburg, 2007) w.r.t. $\mathcal{G}_D$ and *modularity* (Newman, 2006) w.r.t. $\mathcal{G}$ were used as quality metrics for node identity clustering and community detection. We leave details of NCut and modularity in Appendix D.

Logistic regression and $K$Means were used as downstream algorithms for node classification and clustering. Larger F1-score and modularity as well as smaller NCut implies better performance of downstream tasks, thus indicating better embedding quality.

In summary, we adopted (i) *node identity clustering* and (ii) *multi-label node classification* to respectively evaluate identity and position embeddings on the first type of datasets. For the second type of datasets, (i) *multi-label node classification* and (ii) *community detection* were used to evaluate identity and position embeddings. We only applied the unsupervised (i) *node identity clustering* and (ii) *community detection* to evaluate the two types of embeddings for *PPIs*, since we did not consider its ground-truth.

**Baselines**. We compared IRWE with 17 unsupervised baselines, covering identity and position embedding as well as transductive and inductive approaches. Table 3 summarizes all the methods to be evaluated, where '-' denotes that it is unclear for a baseline which type of property it can capture. PhUSION has multiple variants using different proximities for different types of embeddings. We used variants with (i) positive point-wise mutual information and (ii) heat kernel, which are recommended proximities for position and identity embedding, as two baselines denoted as PhN-PPMI and PhN-HK. Each variant of PhUSION can only derive one type of embedding. Although PaCEr also considers the correlation between identity and position embeddings (denoted as PaCEr(I) and PaCEr(P)) and derives both types of embeddings, it only optimizes PaCEr(P) based on the observed graph topology and simply transform PaCEr(P) to PaCEr(I).

For each transductive baseline, we can distinguish that it captures node identities or positions. For inductive baselines, GraLSP, SPINE, and GAS are claimed to be identity embedding methods while P-GNN and CSGCL can preserve node positions. Similar to our method, GraLSP and SPINE use RWs and induced statistics to enhance the embedding quality. SANNE applies the transformer encoder to handle RWs. All the inductive baselines rely on the availability of node attributes. We used the bucket one-hot encodings of node degrees as their attribute inputs, a widely-used strategy for inductive methods when attributes are unavailable. All the transductive methods learn their embeddings only based on graph topology. To validate the challenge of capturing node identities and positions in one embedding space, we introduced an additional baseline [n2v||s2v] by concatenating node2vec and struc2vec.

Most of the baseline can only generate one set of embeddings. We have to use this unique set of embeddings to support two different tasks on each dataset. Our IRWE method can support the inductive inference of identity and position embeddings, simultaneously generating two sets of embeddings. For convenience, we denote the derived identity and position embeddings as **IRWE**($\psi$) and **IRWE**($\gamma$).

As stated in Section 3, we consider the *unsupervised* network embedding. There exist *supervised* inductive methods (e.g., GAT (Veličković et al., 2018), GIN (Xu et al., 2019), ID-GNN (You et al., 2021), DE-GNN (Li et al., 2020), DEMO-Net (Wu et al., 2019), and SAT (Chen et al., 2022)) that do not provide unsupervised training objectives in their original designs. To ensure the fairness of comparison, these supervised baselines are not included in our experiments. Due to space limit, we leave details of layer configurations, parameter settings, and experiment environment in Appendix D.

## 5.2 Evaluation of Transductive Embedding Inference

We first evaluated the transductive embedding inference of all the methods on the first and second types of datasets. For the two classification tasks, we randomly sampled $T \in \{20\%, 40\%, 60\%, 80\%\}$ and 10% of the nodes to form the training and validation sets with the remaining nodes as the test set on each dataset. Similar to 10-fold cross-validation, we repeated the data splitting 10 times, where we split the node set into 10 subsets with each one as the validation set in a round and used the average quality w.r.t. the validation set to tune parameters of all the methods. Evaluation results of the transductive embedding inference are shown in Tables 4 and 5, where metrics are in **bold** or underlined if they perform the best or within top-3.

For transductive baselines, identity embedding approaches (i.e., struc2vec, struc2gauss, and PhN-HK) and position embedding methods (i.e., node2vec, GraRep, PhN-PPMI) are in groups with top clustering performance (in terms of NCut and modularity) on the first and second types of datasets, respectively. Since prior studies have demonstrated the ability of these transductive baselines to capture node identities or positions, the evaluation results *validate our motivation of using node identity clustering and community detection to*

Table 4: Transductive Embedding Inference w.r.t. Node Position Classification and Node Identity Clustering

| | PPI | | | | | Wiki | | | | | BlogCatalog | | | | |
|---|---|---|---|---|---|---|---|---|---|---|---|---|---|---|---|
| | F1-score↑ (%) | | | | Ncut↓ | F1-score↑ (%) | | | | Ncut↓ | F1-score↑ (%) | | | | Ncut↓ |
| | 20% | 40% | 60% | 80% | | 20% | 40% | 60% | 80% | | 20% | 40% | 60% | 80% | |
| node2vec | 17.79 | 19.15 | 20.16 | 21.58 | 45.18 | 47.43 | 51.05 | 52.25 | 53.87 | 38.89 | 37.20 | 39.45 | 40.45 | 41.58 | 36.82 |
| GraRep | 17.94 | 20.54 | 22.00 | 23.49 | 39.92 | 49.87 | 53.33 | 54.18 | 55.09 | 37.12 | 30.83 | 33.58 | 34.71 | 35.68 | 34.32 |
| PaCEr(P) | 15.94 | 17.30 | 18.69 | 19.70 | 45.42 | 43.80 | 45.81 | 46.32 | 47.76 | 36.93 | 35.06 | 37.98 | 38.89 | 39.76 | 34.10 |
| PhN-PPMI | **20.17** | 22.34 | 23.64 | 24.84 | 45.31 | 46.11 | 49.04 | 50.35 | 51.22 | 38.88 | 38.86 | 40.97 | 41.69 | 42.71 | 36.21 |
| struc2vec | 7.70 | 7.99 | 8.04 | 8.47 | 30.51 | 40.70 | 41.14 | 41.17 | 41.34 | 30.96 | 14.67 | 15.09 | 15.28 | 14.79 | 30.47 |
| struc2gauss | 10.59 | 11.40 | 11.91 | 12.59 | 38.01 | 41.09 | 41.06 | 40.86 | 41.13 | 27.66 | 17.16 | 17.21 | 17.28 | 16.95 | 34.41 |
| PaCEr(I) | 9.93 | 10.38 | 10.70 | 10.86 | 40.40 | 41.70 | 41.70 | 41.22 | 42.08 | 24.93 | 16.26 | 16.44 | 16.55 | 16.36 | 32.89 |
| PhN-HK | 9.60 | 9.57 | 9.44 | 9.95 | 31.52 | 41.54 | 41.58 | 41.35 | 41.77 | 29.47 | 17.28 | 17.33 | 17.32 | 17.04 | 34.45 |
| [n2v‖s2v] | 14.29 | 14.67 | 14.66 | 14.38 | 31.99 | 38.95 | 39.75 | 41.85 | 44.37 | 32.32 | 26.94 | 28.75 | 31.34 | 33.75 | 31.14 |
| GraSAGE | 6.59 | 6.29 | 7.12 | 6.88 | 36.00 | 41.14 | 41.06 | 40.82 | 40.89 | 30.71 | 16.79 | 16.77 | 16.70 | 16.56 | 34.28 |
| DGI | 10.98 | 12.37 | 13.36 | 14.24 | 45.35 | 42.63 | 43.44 | 43.91 | 44.33 | 36.85 | 19.24 | 20.81 | 21.92 | 22.22 | 33.35 |
| GraMAE | 11.58 | 12.76 | 13.76 | 14.00 | 37.72 | 42.01 | 42.52 | 42.87 | 43.32 | 25.14 | 19.29 | 20.38 | 20.57 | 21.02 | 28.35 |
| GraMAE2 | 9.63 | 10.40 | 11.26 | 11.52 | 45.26 | 41.85 | 42.04 | 41.73 | 42.34 | 38.26 | 17.76 | 18.14 | 18.23 | 18.29 | 35.56 |
| P-GNN | 11.70 | 12.71 | 13.71 | 13.75 | 39.74 | 43.16 | 44.38 | 44.92 | 45.88 | 37.31 | 19.29 | 20.64 | 21.39 | 21.43 | 34.75 |
| CSGCL | 14.93 | 16.14 | 17.13 | 17.81 | 41.66 | 42.77 | 43.39 | 43.47 | 44.06 | 25.94 | 18.91 | 19.25 | 19.30 | 19.42 | 30.58 |
| GraLSP | 9.08 | 9.35 | 9.37 | 9.95 | 29.76 | 41.05 | 41.00 | 40.62 | 41.40 | 11.00 | 16.65 | 17.50 | 17.44 | 17.58 | **23.46** |
| SPINE | 8.36 | 9.07 | 9.97 | 10.41 | 44.49 | 40.92 | 40.87 | 40.59 | 40.50 | 38.89 | 16.25 | 16.51 | 16.50 | 16.39 | 37.47 |
| GAS | 9.25 | 9.88 | 10.59 | 11.15 | 39.47 | 41.29 | 41.40 | 41.44 | 42.24 | 34.59 | 18.07 | 18.47 | 18.76 | 18.94 | 34.11 |
| SANNE | 7.77 | 8.18 | 8.05 | 9.57 | 46.87 | 41.07 | 41.08 | 41.01 | 41.56 | 38.35 | 16.56 | 16.77 | 16.70 | 16.72 | 37.10 |
| UGFormer | 6.57 | 6.04 | 6.31 | 6.31 | 32.30 | 41.15 | 41.07 | 40.81 | 40.88 | 21.16 | 16.73 | 16.84 | 16.76 | 16.53 | 28.01 |
| **IRWE**($\psi$) | 11.45 | 13.52 | 14.48 | 15.60 | **28.92** | 45.18 | 46.49 | 46.93 | 47.46 | **9.85** | 17.84 | 18.73 | 19.05 | 19.20 | 24.58 |
| **IRWE**($\gamma$) | 19.63 | **22.75** | **24.20** | **25.88** | 42.78 | **52.02** | **54.29** | **54.94** | **56.20** | 19.31 | **38.99** | **41.42** | **41.86** | **42.76** | 36.07 |

Table 5: Transductive Embedding Inference w.r.t. Node Identity Classification and Community Detection

| | USA | | | | | Europe | | | | | Brazil | | | | |
|---|---|---|---|---|---|---|---|---|---|---|---|---|---|---|---|
| | F1-score↑ (%) | | | | Mod↑ (%) | F1-score↑ (%) | | | | Mod↑ (%) | F1-score↑ (%) | | | | Mod↑ (%) |
| | 20% | 40% | 60% | 80% | | 20% | 40% | 60% | 80% | | 20% | 40% | 60% | 80% | |
| node2vec | 47.02 | 50.42 | 53.16 | 53.36 | 25.88 | 36.19 | 39.65 | 41.98 | 41.46 | 7.43 | 32.50 | 32.12 | 39.75 | 37.14 | 11.76 |
| GraRep | 52.52 | 57.86 | 61.93 | 62.01 | 27.54 | 39.18 | 44.32 | 48.09 | 44.87 | 11.48 | 34.89 | 40.45 | 43.50 | 42.14 | 19.76 |
| PaCEr(P) | 47.44 | 49.36 | 51.46 | 53.95 | 22.12 | 42.56 | 45.27 | 48.76 | 50.00 | 4.13 | 37.83 | 39.09 | 45.50 | 50.71 | 2.35 |
| PhN-PPMI | 50.28 | 54.31 | 57.45 | 57.05 | 25.03 | 51.85 | 53.93 | 57.27 | 57.31 | 7.26 | 32.60 | 36.51 | 39.00 | 40.00 | 9.12 |
| struc2vec | 56.85 | 58.97 | 59.91 | 62.52 | 0.38 | 51.85 | 53.93 | 57.27 | 57.31 | -5.61 | 65.43 | 71.66 | 75.25 | 74.29 | -1.43 |
| struc2gauss | **60.88** | 61.89 | 62.32 | 64.36 | 3.27 | 49.50 | 53.38 | 55.53 | 56.34 | -6.49 | 68.69 | 72.72 | **75.50** | 73.57 | -3.31 |
| PaCEr(I) | 59.80 | 60.47 | 60.42 | 61.40 | -0.19 | 50.61 | 54.84 | 56.12 | 59.92 | -3.60 | 63.91 | 67.86 | 68.18 | 73.75 | -2.74 |
| PhN-HK | 58.64 | 60.97 | 62.43 | 63.19 | 13.14 | 50.32 | 54.79 | 54.79 | 56.09 | -6.01 | 61.84 | 68.78 | 74.75 | 69.28 | -5.19 |
| [n2v‖s2v] | 54.02 | 55.69 | 58.79 | 57.05 | 2.91 | 48.25 | 52.23 | 54.79 | 52.43 | -5.22 | 59.78 | 65.75 | 64.75 | 60.71 | 2.28 |
| GraSAGE | 45.49 | 50.06 | 54.70 | 55.37 | 1.55 | 34.23 | 46.31 | 45.70 | 46.82 | -0.71 | 35.86 | 39.09 | 54.00 | 57.85 | 2.93 |
| DGI | 54.62 | 57.78 | 58.85 | 59.49 | 3.45 | 44.23 | 48.05 | 52.39 | 49.02 | -4.78 | 36.19 | 41.36 | 48.25 | 47.85 | 9.18 |
| GraMAE | 58.86 | 62.33 | 64.62 | 64.11 | 5.86 | 45.19 | 49.10 | 52.72 | 49.26 | 1.70 | 44.56 | 55.00 | 63.00 | 66.42 | 3.18 |
| GraMAE2 | 55.91 | 56.90 | 57.67 | 59.07 | 18.73 | 35.97 | 40.09 | 43.96 | 42.92 | 7.03 | 36.63 | 38.93 | 39.00 | 37.85 | 5.95 |
| P-GNN | 58.55 | 61.29 | 62.54 | 61.34 | 21.48 | 45.33 | 47.06 | 51.65 | 50.00 | 0.29 | 46.08 | 50.15 | 49.75 | 52.85 | 1.78 |
| CSGCL | 59.49 | 59.41 | 61.79 | 61.09 | 21.14 | 46.87 | 53.03 | 56.36 | 52.68 | -8.61 | 38.91 | 44.39 | 48.50 | 52.14 | 13.04 |
| GraLSP | 57.89 | 58.87 | 60.58 | 61.84 | 2.72 | 42.59 | 47.66 | 45.70 | 51.70 | 0.65 | 43.15 | 52.12 | 61.25 | 64.28 | 0.32 |
| SPINE | 35.07 | 37.42 | 40.64 | 40.25 | 2.16 | 25.12 | 25.82 | 23.71 | 30.00 | -0.08 | 23.36 | 21.51 | 19.25 | 23.57 | 0.05 |
| GAS | 60.46 | 62.97 | 64.48 | 64.45 | 22.45 | 51.56 | 52.18 | 55.12 | 58.04 | 5.20 | 67.06 | 69.09 | 72.75 | 74.28 | 1.51 |
| SANNE | 54.95 | 56.86 | 58.15 | 61.01 | 14.59 | 44.63 | 50.25 | 54.46 | 49.51 | 6.21 | 40.43 | 45.61 | 51.25 | 51.43 | 5.90 |
| UGFormer | 51.61 | 53.85 | 53.95 | 55.88 | 0.78 | 36.12 | 43.83 | 45.79 | 48.29 | 1.35 | 35.22 | 39.70 | 47.00 | 46.42 | 2.65 |
| **IRWE**($\psi$) | 58.02 | **63.58** | **66.19** | **65.46** | 1.78 | **52.06** | **54.88** | **58.10** | **60.24** | -0.52 | **70.22** | **74.09** | 72.25 | **75.00** | 1.17 |
| **IRWE**($\gamma$) | 55.25 | 58.69 | 60.64 | 61.68 | **31.24** | 43.67 | 47.41 | 50.25 | 49.27 | **17.74** | 36.85 | 40.15 | 44.25 | 41.43 | **21.26** |

*evaluate the quality of identity and position embeddings.* Our node identity clustering results also validate **Hypothesis 1** that *the high-order degree features $\{\delta(v)\}$ can encode node identity information.*

On each dataset, most baselines can only achieve relatively high performance for one task w.r.t. identity or position embedding. It indicates that *most existing embedding methods can only capture either node identities or positions.* In most cases, [n2v‖s2v] outperforms neither (i) node2vec for tasks w.r.t. node positions nor (ii) struc2vec for those w.r.t. node identities. It implies that *the simple integration of the two types of embeddings may even damage the quality of capturing node identities or positions.* Therefore, *it is challenging to preserve both properties in a common embedding space.*

For tasks w.r.t. each type of embedding, conventional transductive baselines can achieve much better performance than most of the advanced inductive baselines. One possible reason is that existing inductive embedding approaches rely on the availability of node attributes. However, there are complicated correlations between graph topology and attributes as discussed in Section 1. Our results imply that *the embedding quality of some inductive baselines is largely affected by their attribute inputs. Some standard settings for the*

Table 6: Inductive Inference for New Nodes within a Graph and across Graphs

| | PPI | | Wiki | | BlogCatalog | | USA | | Europe | | Brazil | | PPIs | |
|---|---|---|---|---|---|---|---|---|---|---|---|---|---|---|
| | F1↑ (%) | Ncut↓ | F1↑ (%) | Ncut↓ | F1↑ (%) | Ncut↓ | F1↑ (%) | Mod↑ (%) | F1↑ (%) | Mod↑ (%) | F1↑ (%) | Mod↑ (%) | Mod↑ (%) | Ncut↓ |
| GraSAGE | 7.35 | 36.13 | 40.71 | 28.86 | 16.30 | **26.50** | 57.81 | 0.88 | 52.68 | 0.02 | 71.42 | 2.18 | 3.90 | 6.69 |
| DGI | 14.64 | 45.18 | 44.16 | 36.89 | 22.76 | 33.71 | 65.54 | 3.90 | 52.19 | 0.69 | 58.57 | 3.65 | 3.52 | 8.31 |
| GraMAE | 14.54 | 38.58 | 43.78 | 24.73 | 20.94 | 27.78 | 66.72 | 1.62 | 54.15 | 1.11 | 64.29 | 3.42 | 2.82 | 7.40 |
| GraMAE2 | 11.81 | 45.28 | 41.27 | 37.88 | 19.05 | 35.94 | 59.83 | 9.56 | 46.34 | 4.46 | 42.86 | 5.34 | 3.68 | 8.14 |
| PGNN | 14.29 | 42.91 | 43.74 | 37.57 | 22.06 | 35.33 | 59.59 | 16.15 | 51.70 | 1.36 | 61.42 | 3.93 | 7.83 | 7.95 |
| CSGCL | 16.13 | 41.46 | 43.96 | 25.54 | 19.30 | 31.14 | 63.36 | 18.17 | 56.59 | -7.81 | 61.43 | 5.81 | -0.26 | 5.88 |
| GraLSP | 6.39 | 47.24 | 40.62 | 31.78 | 16.51 | 37.39 | 25.21 | -0.34 | 44.39 | 0.12 | 38.57 | -0.80 | 0.88 | 8.48 |
| SPINE | 9.12 | 47.21 | 40.80 | 38.95 | 16.87 | 37.45 | 44.87 | 0.76 | 24.88 | 0.16 | 37.14 | 0.41 | 0.38 | 8.63 |
| GAS | 11.50 | 39.33 | 41.84 | 34.44 | 18.94 | 33.89 | 64.87 | 23.05 | 56.59 | 3.51 | 68.57 | 4.27 | -2.10 | 7.15 |
| SANNE | 5.19 | 45.58 | 40.86 | 33.88 | 16.39 | 34.11 | 25.71 | 0.01 | 26.34 | -0.01 | 25.13 | -0.01 | 1.43 | 8.22 |
| UGFormer | 5.59 | 34.70 | 40.71 | 21.39 | 16.23 | 27.84 | 59.83 | 2.03 | 45.85 | 0.73 | 62.86 | 1.95 | -0.83 | 5.43 |
| **IRWE($\psi$)** | 10.53 | **32.95** | 41.05 | **15.93** | 16.10 | 26.58 | **68.40** | 10.07 | **60.00** | -1.12 | **72.86** | -5.28 | 0.16 | **4.62** |
| **IRWE($\gamma$)** | **18.29** | 45.54 | **47.32** | 19.47 | **27.04** | 35.96 | 49.75 | **25.54** | 45.85 | **11.65** | 44.29 | **12.31** | **11.41** | 8.47 |

*case without available attributes (e.g., using one-hot degree encodings as attribute inputs) cannot help derive informative identity or position embeddings.*

Our IRWE method achieves the best quality for both identity and position embedding in most cases. It indicates that *IRWE can jointly derive informative identity and position embeddings in a unified framework.*

## 5.3 Evaluation of Inductive Embedding Inference

We further consider the inductive inference (i) *for new unseen nodes within a graph* and (ii) *across graphs*, which were evaluated on the (i) first two types of datasets (i.e., *PPI*, *Wiki*, *BlogCatalog*, *USA*, *Europe*, and *Brazil*) and (ii) *PPIs*, respectively. We could only evaluate the quality of inductive methods because transductive baselines cannot support the inductive inference.

For the *inductive inference within a graph*, we randomly selected 80%, 10%, and 10% of nodes on each single graph to form the training, validation, and test sets (denoted as $\mathcal{V}_{trn}$, $\mathcal{V}_{val}$, and $\mathcal{V}_{tst}$), where $\mathcal{V}_{val}$ and $\mathcal{V}_{tst}$ represent sets of new nodes not observed in $\mathcal{V}_{trn}$. The embedding model of each inductive method was optimized only on the topology induced by $\mathcal{V}_{trn}$. When validating and testing a method using the node classification task, embeddings w.r.t. $\mathcal{V}_{trn}$ and $\mathcal{V}_{trn} \cup \mathcal{V}_{val}$ were used to train the downstream logistic regression. We repeated the data splitting 10 times following a strategy similar to 10-fold cross validation and used the average quality w.r.t. the validation set to tune parameters of all the methods.

For the *inductive inference across graphs*, we sampled 3 graphs from *PPIs* denoted as $\mathcal{G}_{trn}$, $\mathcal{G}_{val}$, and $\mathcal{G}_{tst}$, which were used for training, validation, and testing. We first optimized the embedding model on $\mathcal{G}_{trn}$. To validate or test the model, we derived inductive embeddings w.r.t. $\mathcal{G}_{val}$ or $\mathcal{G}_{tst}$ and obtained clustering results for evaluation by applying *K*Means. This procedure was repeated 5 times, where 15 graphs were sampled. Finally, the average quality over the 5 data splits was reported.

Evaluation results of the inductive embedding inference are depicted in Table 6, where metrics are in **bold** or underlined if they perform the best or within top-3. IRWE achieves the best quality in most cases. In particular, the quality metrics of IRWE are significantly better than other inductive baselines, whose inductiveness relies on the availability of node attributes. Our results further demonstrate that *IRWE can support the inductive inference of identity and position embeddings, simultaneously generating two sets of informative embeddings, without relying on the availability and aggregation of any graph attributes.*

## 5.4 Ablation Study

In ablation study, we respectively removed some components from the IRWE model to explore their effectiveness for ensuring the high embedding quality of our method. For the *identity embedding module*, we considered the (i) AW embedding regularization loss $\mathcal{L}_{\text{reg}-\varphi}$ (12), (ii) AW statistic inputs $\{\tilde{s}(v)\}$, (iii) high-order degree feature inputs $\{\delta(v)\}$, and (iv) identity embedding regularization loss $\mathcal{L}_{\text{reg}-\psi}$ (13). In cases (i) and (iv), identity embeddings were only optimized via one loss (i.e., $\mathcal{L}_{\text{reg}-\psi}$ or $\mathcal{L}_{\text{reg}-\varphi}$).

Table 7: Ablation Study w.r.t. Node Position Classification and Node Identity Clustering on *PPI* as well as Node Identity Classification and Community Detection on *USA*.

|  | *PPI* | | *USA* | |
|---|---|---|---|---|
|  | **F1**↑ (%) | **Ncut**↓ | **F1**↑ (%) | **Mod**↑ (%) |
| **IRWE** | **25.88** | **28.94** | **67.31** | **31.24** |
| (1) w/o loss $\mathcal{L}_{\text{reg}-\varphi}$ | 25.43 | 30.14 | 66.55 | 30.82 |
| (2) w/o input $\{\tilde{s}(v)\}$ | 24.76 | 29.68 | 65.21 | 29.31 |
| (3) w/o input $\{\delta(v)\}$ | 25.14 | 30.61 | 67.07 | 31.08 |
| (4) w/o loss $\mathcal{L}_{\text{reg}-\psi}$ | 25.65 | 36.02 | 45.79 | 30.11 |
| (5) w/o input $\{\psi(v)\}$ | 24.95 | 29.28 | 65.79 | 30.44 |
| (6) w/o input $\{\pi_g(v)\}$ | 25.08 | 29.62 | 65.79 | 29.45 |
| (7) w/o ROut($\cdot$) | 13.39 | 29.39 | 66.47 | -0.76 |
| (8) w/o loss $\mathcal{L}_\gamma$ | 22.43 | 29.42 | 65.88 | 23.65 |
| (9) base stat $\{\tilde{s}(v)\}$ | – | 46.05 | 56.63 | – |
| (10) base stat $\{\delta(v)\}$ | – | 34.06 | 63.94 | – |
| (11) base stat $\{\pi_g(v)\}$ | 17.52 | – | – | 21.85 |
| (12) based stat **C** (SVD) | 22.60 | – | – | 12.15 |

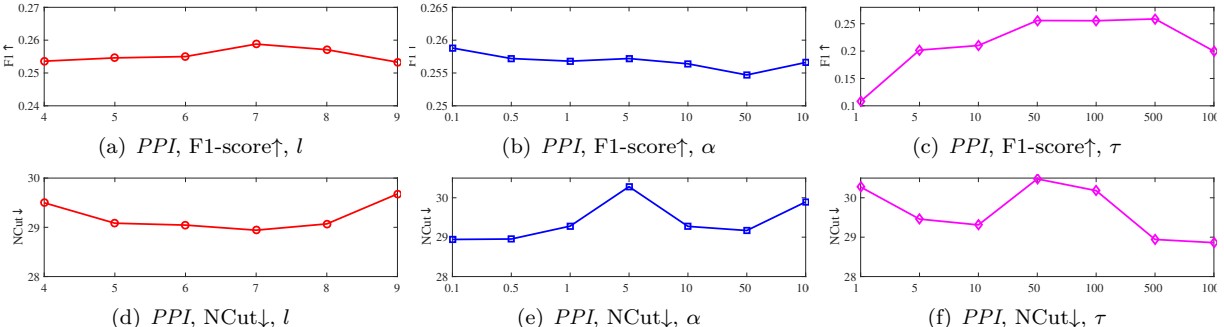

(a) *PPI*, F1-score↑, $l$  (b) *PPI*, F1-score↑, $\alpha$  (c) *PPI*, F1-score↑, $\tau$

(d) *PPI*, NCut↓, $l$  (e) *PPI*, NCut↓, $\alpha$  (f) *PPI*, NCut↓, $\tau$

Figure 5: Parameter analysis w.r.t. $l$, $\alpha$, and $\tau$ on *PPI* in terms of F1-score↑ (node position classification) and NCut↓ (node identity clustering).

For the *position embedding module*, we checked the effectiveness of the (v) identity embedding inputs $\{\psi(v)\}$, (vi) global position encoding inputs $\{\pi_g(v)\}$, (vii) attentive readout function ROut($\cdot$) described in (9), and (viii) reconstruction loss $\mathcal{L}_\gamma$ (14). In case (v), the two modules of IRWE were independently optimized. For case (vii), we simply averaged the representations in $\bar{t}^{(v)}$ to replace ROut($\cdot$). For case (viii), we replaced the contrastive statistics **C** in (14) with adjacency matrix **A**.

We also used some induced statistics as baselines. Concretely, we evaluated the quality of (ix) AW statistics $\{\tilde{s}(v)\}$ and (x) degree features $\{\delta(v)\}$ to capture node identities. In contrast, we checked the quality of (xi) global position encodings $\{\pi_g(v)\}$ and (xii) contrastive statistics **C** for node positions. In case (xii), we derived representations with the same dimensionality as other embedding methods by applying SVD to **C**.

As a demonstration, we report results of transductive embedding inference on *PPI* and *USA* (with 80% of nodes sampled as the training set for classification) in Table 7. According to our results, $\mathcal{L}_{\text{reg}-\psi}$ *is essential for identity embedding learning*, since there are significant quality declines for node identity clustering and classification in case (iv). ROut($\cdot$) *and* $\mathcal{L}_\gamma$ *are key components to capture node positions* due to the significant quality declines for node position classification and community detection in cases (vii) and (viii). *All the remaining components can further enhance the ability to capture node identities and positions. The joint optimization of identity and position embeddings can also improve the quality of one another.*

## 5.5 Parameter Analysis

We tested the effects of (i) RW length $l$, (ii) $\alpha$ in loss (11), and (iii) temperature parameter $\tau$ in loss (14). Concretely, we set $l \in \{4, 5, \cdots, 9\}$, $\alpha \in \{0.1, 0.5, 1, 5, 10, 50, 100\}$, and $\tau \in \{1, 5, 10, 50, 100, 500, 1000\}$. Example parameter analysis results of the transductive embedding inference on *PPI* and *USA* (with 80% of nodes sampled as the training set for classification) are illustrated in Fig. 5 and 6. The quality of both types of embeddings is not sensitive to the settings of $l$. Compared with position embeddings, the quality

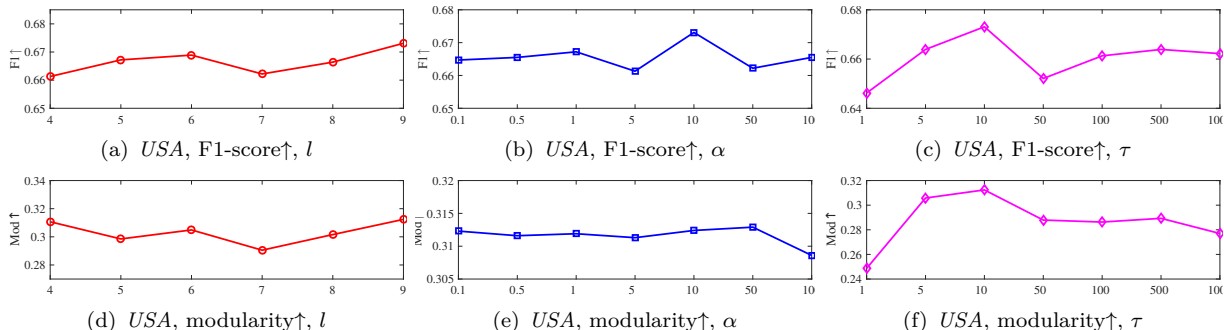

Figure 6: Parameter analysis w.r.t. $l$, $\alpha$, and $\tau$ on *USA* in terms of F1-score↑ (node identity classification) and modularity↑ (community detection).

of identity embeddings is more sensitive to $\alpha$ (e.g., in terms of F1-score of node classification on *USA* and NCut of node identity clustering on *PPI*). The settings of $\tau$ would significantly affect the quality of the two types of embeddings. The recommended parameter settings of IRWE are given in the Appendix D.

## 6 Conclusion

In this paper, we considered unsupervised network embedding and explored a unified framework for the joint optimization and inductive inference of identity and position embeddings without relying on the availability and aggregation of graph attributes. An IRWE method was proposed, which combines multiple attention units with different designs to handle RWs on graph topology. We demonstrated that AW derived from RW and induced statistics can (i) be features shared by all possible nodes and graphs to support inductive inference and (ii) characterize node identities to derive identity embeddings. We also showed the intrinsic relation between the two types of embeddings. Based on this relation, the derived identity embeddings can be used for the inductive inference of position embeddings. Experiments on public datasets validated that IRWE can achieve superior quality compared with various baselines for the transductive and inductive inference of identity and position embeddings. We leave discussions of future directions in Appendix F.

### Acknowledgments

This research has been made possible by funding support provided to Dit-Yan Yeung by the Research Grants Council of Hong Kong under the Research Impact Fund project R6003-21.

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

## A  Detailed Algorithms

The RW sampling procedure starting from a node is summarized in Algorithm 6, which uniformly sample the next node $v_t$ from the neighbors of each source node $v_s$.

Algorithm 7 summarizes the *transductive inference procedure* of IRWE, where the RWs $\{\mathcal{W}^{(v)}, \mathcal{W}_I^{(v)}\}$, AW lookup table $\tilde{\Omega}_l$, and statistics $\{\tilde{s}(v), \delta(v), \pi_g(v)\}$ derived the model optimization are used as the inputs.

---

**Algorithm 6:** RW Sampling Starting from a Node

---

**Input:** topology $(\mathcal{V}, \mathcal{E})$; target node $v$; RW length $l$; number of samples $n_S$
**Output:** set of sampled RWs $\mathcal{W}^{(v)}$

**1** $\mathcal{W}^{(v)} \leftarrow \emptyset$ //Initialize $\mathcal{W}^{(v)}$
**2** **for** *sample_count* **from** 1 **to** $n_S$ **do**
**3** $\quad$ $v_s \leftarrow v$ and $w \leftarrow (v_s)$ //Initialize current RW $w$
**4** $\quad$ **while** $|w| \leq (l+1)$ **do**
**5** $\quad\quad$ randomly sample a node $v_t$ from $v_s$'s neighbors
**6** $\quad\quad$ append $v_t$ to $w$
**7** $\quad\quad$ $v_s \leftarrow v_t$
**8** $\quad$ add $w$ to $\mathcal{W}^{(v)}$

---

---

**Algorithm 7:** Transductive Inference

---

**Input:** RWs $\{\mathcal{W}^{(v)}, \mathcal{W}_I^{(v)}\}$, AW lookup table $\tilde{\Omega}_l$, & statistics $\{\tilde{s}(v), \delta(v), \pi_g(v)\}$ saved in model optimization;
$\quad\quad\quad$ inference topology $(\mathcal{V}, \mathcal{E})$
**Output:** *transductive* embddings $\{\psi(v)\}$ & $\{\gamma(v)\}$ w.r.t. $\mathcal{V}$

**1** get $\{\psi(v)\}$ based on $\{\tilde{\Omega}_l, \tilde{s}(v), \delta(v)\}$ w.r.t. $\mathcal{V}$
**2** get $\{\gamma(v)\}$ based on $\{\psi(v), \pi_g(v), \pi_l(j), \mathcal{W}_I^{(v)}\}$ w.r.t. $\mathcal{V}$

---

Therefore, the *transudcitve inference* of identity embeddings $\{\psi(v)\}$ and position embeddings $\{\gamma(v)\}$ only includes one feedforward propagation through the model.

Procedures to get *inductive* AW statistics $\{s(v)\}$, high-order degree features $\{\delta(v)\}$, and global position encodings $\{\pi_g(v)\}$, which support the *inductive inference for new nodes within a graph* (i.e., Algorithm 5), are described in Algorithms 8, 9, and 10, respectively. When deriving $\{s(v)\}$, we only compute the frequency of AWs in the lookup table $\tilde{\Omega}_l$ reduced on $(\mathcal{V}, \mathcal{E})$ rather than all AWs. Moreover, we get $\{\delta(v)\}$ based on the one-hot degree encoding truncated by the minimum and maximum degrees of the training topology $(\mathcal{V}, \mathcal{E})$ but not those of the inference topology $(\mathcal{V} \cup \mathcal{V}', \mathcal{E}')$. For $\pi_g(v)$, we compute truncated RW statistic $r(v)$ only w.r.t. previously observed nodes $\mathcal{V}$ rather than $\mathcal{V}' \cup \mathcal{V}$.

The *inductive inference across graphs* is summarized in Algorithm 11. We sample RWs $\{\mathcal{W}^{(v)}, \mathcal{W}_I^{(v)}\}$ on each new graph $(\mathcal{V}'', \mathcal{E}'')$. Since there are no shared nodes between the training topology $(\mathcal{V}, \mathcal{E})$ and inference topology $(\mathcal{V}'', \mathcal{E}'')$, we only incrementally compute statistics $\{\tilde{s}(v), \delta(v)\}$ based on $\{\tilde{\Omega}_l, \deg_{\min}, \deg_{\max}\}$ derived from $(\mathcal{V}, \mathcal{E})$ but compute global position encodings $\{\pi_g(v)\}$ from scratch.

## B  Complexity Analysis

The complexity of the RW sampling starting from each node (i.e., Algorithm 6) is no more than $O(n_S l)$. The complexities to derive AW statistics $s(v)$ (i.e., Algorithm 1), high-order degree features $\delta(v)$ (i.e., Algorithm 2), and global position encoding $\pi_g(v)$ (i.e., Algorithm 3) w.r.t. a node $v$ are $O(n_S)$, $O(n_S l)$, and $O(n_S l + k(v)d)$, with $k(v)$ as the number of nodes observed in $\mathcal{W}^{(v)}$. The overall complexity to derive the RW-induced statistics (i.e., the feature inputs of IRWE) from a graph $(\mathcal{V}, \mathcal{E})$ is no more than $O(|\mathcal{V}|n_S l + |\mathcal{V}|n_S + |\mathcal{V}|n_S l + (|\mathcal{V}|n_S l + \bar{k}d)) = O(|\mathcal{V}|n_S l + \bar{k}d)$, with $\bar{k} := \sum_{v \in \mathcal{V}} k(v)$.

As described in Algorithm 7, the *transductive inference* of IRWE only includes one feedforward propagation through the model. Its complexity is no more than $O(\tilde{\eta}_l l^2 d + |\mathcal{V}|(el + \tilde{\eta}_l)d + |\mathcal{V}|\tilde{\eta}_l dh + (|\mathcal{V}|d^2 + |\mathcal{V}|d) + |\mathcal{V}|(d + l)d + |\mathcal{V}|n_I l^2 dh + n_I d) = O(|\mathcal{V}|(\tilde{\eta}_l + n_I l^2)dh)$, where we assume that $el \approx d$, $l^2 \ll |\mathcal{V}|$, and $d \ll \tilde{\eta}_l$; $h$ is the number of attention heads. According to Algorithm 5, the complexity of *inductive inference for new nodes within a graph* is $O(|\mathcal{V}'|n_S l + \bar{k}'d + |\mathcal{V} \cup \mathcal{V}'|(\tilde{\eta}_l + n_I l^2)dh)$, with $\bar{k}' := \sum_{v \in \mathcal{V}'} k(v)$. The complexity of *inductive inference across graphs* (i.e., Algorithm 11) is $O(|\mathcal{V}''|n_S l + \bar{k}''d + |\mathcal{V}''|(\tilde{\eta}_l + n_I l^2)dh)$, with $\bar{k}'' := \sum_{v \in \mathcal{V}''} k(v)$.

Table 8 summarizes and compares the complexities of model parameters to be learned for all the methods in our experiments, where $N$ is the number of nodes; $d$ is the dimensionality of input feature or embedding;

---

**Algorithm 8:** Inductive Derivation of AW Statistics

---

**Input:** new target node $v \in \mathcal{V}'$; sampled RWs $\mathcal{W}^{(v)}$; AW lookup table $\tilde{\Omega}_l$ reduced on old topology $(\mathcal{V}, \mathcal{E})$

**Output:** *inductive* AW statistic $s(v)$ w.r.t. $v$

**1** $\tilde{\eta}_l \leftarrow |\tilde{\Omega}_l|$ //Get size of reduced AW lookup table

**2** $s(v) \leftarrow [0, 0, \cdots, 0]^{\tilde{\eta}_l}$ //Initialize $s(v)$

**3 for each** $w \in \mathcal{W}^{(v)}$ **do**

**4**     map RW $w$ to its corresponding AW $\omega$

**5**     **if** $\omega \in \tilde{\Omega}_l$ **then**

**6**        get the index $j$ of AW $\omega$ in reduced lookup table $\tilde{\Omega}_l$

**7**        $s(v)_j \leftarrow s(v)_j + 1$ //Update $s(v)$

---

**Algorithm 9:** Inductive Derivation of Degree Feature

---

**Input:** new target node $v \in \mathcal{V}'$; RW length $l$; one-hot degree encoding dimensionality $e$; sampled RWs $\mathcal{W}^{(v)}$;    $\deg_{\min}$ & $\deg_{\max}$ in old topology $(\mathcal{V}, \mathcal{E})$

**Output:** *inductive* degree feature $\delta(v)$ w.r.t. $v$

**1** $\delta(v) \leftarrow [0, 0, \cdots, 0]^{(l+1)e}$ //Initialize degree feature $\delta(v)$

**2 for each** $w \in \mathcal{W}^{(v)}$ **do**

**3**     **for** $i$ **from** $0$ **to** $l$ **do**

**4**        $u \leftarrow w^{(i)}$ //$i$-th node in current RW $w$

**5**        **if** $u \in \mathcal{V}$ **then**

**6**           $\rho_d(u) \leftarrow [0, \cdots, 0]^e$ //Initialize one-hot degree encoding

**7**           **if** $\deg(u) < \deg_{\min}$ **then**

**8**              $j \leftarrow 0$

**9**           **else if** $\deg(u) > \deg_{\max}$ **then**

**10**             $j \leftarrow (e-1)$

**11**           **else**

**12**             $j \leftarrow \left\lfloor \frac{(\deg(u) - \deg_{\min})e}{(\deg_{\max} - \deg_{\min})} \right\rfloor$

**13**           $\rho_d(u)_j \leftarrow 1$ //Update $\rho_d(u)$

**14**           $\delta(v)_{ie:(i+1)e} \leftarrow \delta(v)_{ie:(i+1)e} + \rho_d(u)$

---

$l$ is the RW/AW length; $\eta_l$ and $\tilde{\eta}_l$ denote the (i) number of AWs w.r.t. length $l$ and (ii) reduced value of $\eta_l$; $L$ is the number of layers of GNN or transformer encoder; $h$ is the number of attention heads. Since most transductive embedding methods (e.g., *node2vec* and *struc2vec*) follow the embedding lookup scheme, their model parameters are with a complexity of at least $O(Nd)$. Most inductive approaches rely on the attribute aggregation mechanism of GNNs. Their model parameters have a complexity of at least $O(Ld^2)$. Methods based on the multi-head attention or transformer encoder (e.g., *SANNE*, *UGFormer*, and **IRWE**) should have at least $O(Lhd^2)$ learnable model parameters. In addition, IRWE also includes the AW auto-encoder and MLPs in the identity embedding encoder and decoder. Therefore, the model parameters of IRWE have a complexity of $O(l^2d + (\tilde{\eta}_l + le)d + Lhd^2) = O(l^2d + \tilde{\eta}_l d + Lhd^2)$, where we assume that $el \approx d$.

## C    Proof of Proposition 1

For simplicity, we let $z_{ij} := \gamma(v_i)\tilde{\gamma}^T(v_j)/\tau$. To minimize the contrastive loss $\mathcal{L}_{\text{cnr}}$ (10), one can let its partial derivative $\partial\mathcal{L}_{\text{cnr}}/z_{ij}$ w.r.t. each edge $(v_i, v_j) \in \mathcal{E}$ to 0. Since $\sigma(x) = 1/(1 + e^{-x})$ and $d\sigma(x)/dx = \sigma(x)[1 - \sigma(x)]$, we have

$$0 = \partial\mathcal{L}_{cnr}/z_{ij} = -p_{ij}(1 - \sigma(z_{ij})) + Qn_j(1 - \sigma(-z_{ij})), \tag{15}$$

which can be rearranged as

$$p_{ij}\sigma(z_{ij}) - Qn_j\sigma(-z_{ij}) = p_{ij} - Qn_j. \tag{16}$$

---

**Algorithm 10:** Inductive Derivation of Global Position Encoding

---

**Input:** new target node $v \in \mathcal{V}'$; sampled RWs $\mathcal{W}^{(v)}$; old training node set $\mathcal{V}$; random matrix $\mathbf{\Theta} \in \mathbb{R}^{|\mathcal{V}| \times d}$

**Output:** *inductive* global position encoding $\pi_g(v)$ w.r.t. $v$

1   $r(v) \leftarrow [0, 0, \cdots, 0]^{|\mathcal{V}|}$ //Initialize RW stat $r(v)$

2   **for each** $w \in \mathcal{W}^{(v)}$ **do**

3     **for each** *node* $v \in w$ **do**

4       **if** $v \in \mathcal{V}$ **then**

5         get index $j$ of $v$ in the training node set $\mathcal{V}$

6         $r(v)_j \leftarrow r(v)_j + 1$ //Update $r(v)$

7   $\pi_g(v) \leftarrow r(v)\mathbf{\Theta}$ //Derive $\pi_g(v)$

---

**Algorithm 11:** Inductive Inference across Graphs

---

**Input:** optimized model parameters $\{\theta_\psi^*, \theta_\gamma^*\}$; new topology $(\mathcal{V}'', \mathcal{E}'')$; RW settings $\{l, n_S, n_I\}$; local position encodings $\{\pi_l(j)\}$; $\{\tilde{\Omega}_l, \deg_{\min}, \deg_{\max}\}$ derived in model optimization on old topology $(\mathcal{V}, \mathcal{E})$

**Output:** *inductive* embeddings $\{\psi(v)\}$ & $\{\gamma(v)\}$ w.r.t. $\mathcal{V}''$

1   **for each** *node* $v \in \mathcal{V}''$ **do**

2     sample $n_S$ RWs $\mathcal{W}^{(v)}$ from $v$ w.r.t. $\mathcal{E}''$ via Algorithm 6

3     get AW statistics $\tilde{s}(v)$ w.r.t. $\{\mathcal{W}^{(v)}, \tilde{\Omega}_l\}$ via Algorithm 8

4     get degree feature $\delta(v)$ w.r.t. $\{\mathcal{W}^{(v)}, d_{\min}, d_{\max}\}$ via Algorithm 9

5     get global position encoding $\pi_g(v)$ w.r.t. $\mathcal{W}^{(v)}$ via Algorithm 3

6     randomly select $n_I$ RWs $\mathcal{W}_I^{(v)}$ from $\mathcal{W}^{(v)}$

7   get $\{\psi(v)\}$ based on $\{\tilde{\Omega}_l, \tilde{s}(v), \delta(v)\}$ w.r.t. $\mathcal{V}''$

8   get $\{\gamma(v)\}$ based on $\{\psi(v), \pi_g(v), \pi_l(j), \mathcal{W}_I^{(v)}\}$ w.r.t. $\mathcal{V}''$

---

Table 8: Summary of the Complexities of Model Parameters to be Learned.

| node2vec | GraRep | struc2vec | struc2gauss | PaCEr | PhN | GSAGE | DGI | GMAE |
|---|---|---|---|---|---|---|---|---|
| $O(Nd)$ | $O(Ndl)$ | $O(Nd)$ | $O(Nd)$ | $O(N(d+N))$ | $O(Nd)$ | $O(Ld^2)$ | $O(Ld^2)$ | $O(Ld^2)$ |
| GMAE2 | P-GNN | CSGCL | GraLSP | SPINE | GAS | SANNE | UGFormer | **IRWE** |
| $O(Ld^2)$ | $O(Ld^2)$ | $O(Ld^2)$ | $O(Ld^2 + \eta_l d)$ | $O(d^2)$ | $O(Ld^2)$ | $O(Lhd^2)$ | $O(Lhd^2)$ | $O(l^2 d + \tilde{\eta}_l d + Lhd^2)$ |

By applying $\sigma(-x) = e^{-x}\sigma(x)$, we have

$$
\begin{aligned}
&p_{ij}\sigma(z_{ij}) - Qn_j \cdot \exp\{-z_{ij}\}\sigma(z_{ij}) = p_{ij} - Qn_j \\
&\Rightarrow \frac{p_{ij} - Qn_j \cdot \exp\{-z_{ij}\}}{1 + \exp\{-z_{ij}\}} = p_{ij} - Qn_j \\
&\Rightarrow \frac{p_{ij} + Qn_j - Qn_j(1 + \exp\{-z_{ij}\})}{1 + \exp\{-z_{ij}\}} = p_{ij} - Qn_j \\
&\Rightarrow (p_{ij} + Qn_j)\sigma(z_{ij}) = p_{ij} \\
&\Rightarrow \sigma(z_{ij}) = p_{ij}/(p_{ij} + Qn_j) \\
&\Rightarrow 1 + \exp\{-z_{ij}\} = (p_{ij} + Qn_j)/p_{ij} \\
&\Rightarrow \exp\{-z_{ij}\} = Qn_j/p_{ij}
\end{aligned}
\tag{17}
$$

By taking the logarithm of both sides, we further have

$$
z_{ij} = \ln p_{ij} - \ln(Qn_j).
\tag{18}
$$

Let $\mathbf{C} \in \mathbb{R}^{|\mathcal{V}| \times |\mathcal{V}|}$ be an auxiliary matrix with the same definition as that in **Proposition 1**. From the perspective of matrix factorization, we can rewrite the aforementioned equation to another matrix form $\mathbf{\Gamma}\tilde{\mathbf{\Gamma}}^T/\tau = \mathbf{C}$, which is equivalent to the reconstruction loss $\mathcal{L}_\gamma$ (14).

Table 9: Parameter Settings for Transductive Inference.

| | $(d, e, n_S, n_I)$ | $(\lambda_\psi, \lambda_\gamma)$ | $(m, m_\psi, m_\gamma)$ | $(l, \alpha, \tau)$ |
|---|---|---|---|---|
| *PPI* | (256, 100, 1e3, 10) | (5e-4,1e-3) | (2e3, 10, 1) | (7, 0.1, 5e2) |
| *Wiki* | (256, 100, 1e3, 10) | (1e-3,1e-3) | (1e3, 5, 1) | (7, 10, 1e3) |
| *Blog* | (256, 100, 1e3, 10) | (5e-4,5e-4) | (3e3, 1, 20) | (9, 10, 10) |
| *USA* | (128, 100, 1e3, 20) | (1e-3,5e-4) | (500, 10, 1) | (9, 10, 10) |
| *Europe* | (64, 100, 1e3, 20) | (5e-4,5e-4) | (200, 1, 1) | (9, 10, 10) |
| *Brazil* | (64, 32, 1e3, 20) | (5e-4,5e-4) | (200, 1, 1) | (9, 0.1, 1e2) |

Table 10: Parameter Settings for Inductive Inference.

| | $(d, e, n_S, n_I)$ | $(\lambda_\psi, \lambda_\gamma)$ | $(m, m_\psi, m_\gamma)$ | $(l, \alpha, \tau)$ |
|---|---|---|---|---|
| *PPI* | (256, 100, 1e3, 10) | (5e-4,1e-4) | (1e3, 20, 1) | (7, 10, 5e2) |
| *Wiki* | (256, 100, 1e3, 10) | (1e-3,5e-4) | (1e3, 1, 1) | (7, 10, 5e2) |
| *Blog* | (256, 100, 1e3, 10) | (5e-4,5e-4) | (1e3, 20, 5) | (5, 10, 5) |
| *USA* | (128, 100, 1e3, 10) | (5e-4,5e-4) | (500, 10, 1) | (9, 10, 10) |
| *Europe* | (64, 100, 1e3, 10) | (5e-4,5e-4) | (200, 1, 1) | (9, 10, 10) |
| *Brazil* | (64, 32, 1e3, 10) | (5e-4,5e-4) | (200, 1, 1) | (9, 0.1, 1e2) |
| *PPIs* | (256, 100, 1e3, 10) | (5e-4,5e-4) | (1000, 5, 1) | (9, 10, 50) |

Table 11: Layer Configurations for Transductive Inference.

| Datasets | Identity Embedding Module | | | | | Position Embedding Module | | |
|---|---|---|---|---|---|---|---|---|
| | $\mathrm{Enc}_\varphi(\cdot)$ | $\mathrm{Dec}_\varphi(\cdot)$ | $\mathrm{Red}_s(\cdot)$ | $h_\psi$ | $\mathrm{Dec}_\psi(\cdot)$ | MLP in $\mathrm{ReAtt}(\cdot)$ | $(L_{\mathrm{tran}}, h_{\mathrm{tran}})$ | $h_{\mathrm{rout}}$ |
| *PPI* | $l^2$,128,t,d,t | d,128,t,$l^2$,t | $\bar{\eta}_l$+le,2048,r,1024,r,512,r,d,r | 64 | d,512,t,le,t | d,d,s,d,s,d,s,d,s | (4, 64) | 64 |
| *Wiki* | $l^2$,128,t,d,t | d,128,t,$l^2$,t | $\bar{\eta}_l$+le,1024,r,512,r,d,r | 64 | d,512,t,le,t | d,d,s,d,s,d,s,d,s | (4, 64) | 64 |
| *Blog* | $l^2$,128,t,d,t | d,128,t,$l^2$,t | $\bar{\eta}_l$+le,1024,r,512,r,d,r | 64 | d,512,t,le,t | d,d,s,d,s,d,s,d,s | (5, 64) | 64 |
| *USA* | $l^2$,100,t,d,t | d,100,t,$l^2$,t | $\bar{\eta}_l$+le,4096,r,2048,r,512,r,d,r | 32 | d,le,t | d,d,s,d,s,d,s,d,s | (4, 32) | 32 |
| *Europe* | $l^2$,64,t,d,t | d,64,t,$l^2$,t | $\bar{\eta}_l$+le,4096,r,1024,r,256,r,d,r | 16 | d,256,t,512,t,le,t | d,d,s,d,s | (4, 16) | 16 |
| *Brazil* | $l^2$,64,t,d,t | d,64,t,$l^2$,t | $\bar{\eta}_l$+le,1024,r,512,r,128,r,d,r | 16 | d,128,t,le,t | d,d,s,d,s | (4, 16) | 16 |

Table 12: Layer Configurations for Inductive Inference.

| Datasets | Identity Embedding Module | | | | | Position Embedding Module | | |
|---|---|---|---|---|---|---|---|---|
| | $\mathrm{Enc}_\varphi(\cdot)$ | $\mathrm{Dec}_\varphi(\cdot)$ | $\mathrm{Red}_s(\cdot)$ | $h_\psi$ | $\mathrm{Dec}_\psi(\cdot)$ | MLP in $\mathrm{ReAtt}(\cdot)$ | $(L_{\mathrm{tran}}, h_{\mathrm{tran}})$ | $h_{\mathrm{rout}}$ |
| *PPI* | $l^2$,128,t,d,t | d,128,t,$l^2$,t | $\bar{\eta}_l$+le,1024,r,512,r,d,r | 64 | d,le,t | d,d,s,d,s,d,s,d,s | (4, 64) | 64 |
| *Wiki* | $l^2$,128,t,d,t | d,128,t,$l^2$,t | $\bar{\eta}_l$+le,1024,r,512,r,d,r | 64 | d,512,t,le,t | d,d,s,d,s,d,s,d,s | (4, 64) | 64 |
| *Blog* | $l^2$,128,t,d,t | d,128,t,$l^2$,t | $\bar{\eta}_l$+le,1024,r,512,r,d,r | 64 | d,t,512,t,le,t | d,d,s,d,s,d,s,d,s | (5, 64) | 64 |
| *USA* | $l^2$,100,t,d,t | d,100,t,$l^2$,t | $\bar{\eta}_l$+le,1024,r,512,r,d,r | 32 | d,512,t,le,t | d,d,s,d,s,d,s,d,s | (4, 16) | 16 |
| *Europe* | $l^2$,64,t,d,t | d,64,t,$l^2$,t | $\bar{\eta}_l$+le,1024,r,512,r,d,r | 16 | d,256,t,512,t,le,t | d,d,s,d,s | (4, 16) | 16 |
| *Brazil* | $l^2$,64,t,d,t | d,64,t,$l^2$,t | $\bar{\eta}_l$+le,512,r,128,r,d | 16 | d,256,t,le,t | d,d,s,d,s | (4, 16) | 16 |
| *PPIs* | $l^2$,128,t,d,t | d,128,t,$l^2$,t | $\bar{\eta}_l$+le,1024,r,512,r,d,r | 64 | d,512,t,le,t | d,d,s,d,s,d,s,d,s | (6, 64) | 64 |

## D  Detailed Experiment Settings

Given a clustering result $\mathcal{C} = \{\mathcal{C}_1, \cdots, \mathcal{C}_K\}$, *NCut* w.r.t. the auxiliary similarity graph $\mathcal{G}_D$ is defined as

$$\mathrm{NCut}(\mathcal{C}; \mathcal{G}_D) := 0.5 \sum_{r=1}^{K} [\mathrm{cut}(\mathcal{C}_r, \bar{\mathcal{C}}_r)/\mathrm{vol}(\mathcal{C}_r)], \tag{19}$$

where $\bar{\mathcal{C}}_r := \mathcal{V} - \mathcal{C}_r$, $\mathrm{cut}(\mathcal{C}_r, \bar{\mathcal{C}}_r) := \sum_{v_i \in \mathcal{C}_r, v_j \in \bar{\mathcal{C}}_r} (\mathbf{A}_D)_{ij}$, and $\mathrm{vol}(\mathcal{C}_r) := \sum_{v_i \in \mathcal{C}_r, v_j \in \mathcal{V}} (\mathbf{A}_D)_{ij}$, with $\mathbf{A}_D$ as the adjacency matrix of $\mathcal{G}_D$. Given a clustering result $\mathcal{C}$, *modularity* w.r.t. the original graph $\mathcal{G}$ is defined as

$$\mathrm{Mod}(\mathcal{C}; \mathcal{G}) := \frac{1}{2e} \sum_{r=1}^{K} \sum_{v_i, v_j \in \mathcal{C}_r} [\mathbf{A}_{ij} - \deg(v_i)\deg(v_j)/(2e)], \tag{20}$$

where $e := \sum_i \deg(v_i)/2$ is the number of edges.

The parameter settings of IRWE for the *transductive* and *inductive* inference are depicted in Tables 9 and 10, where $d$ is the embedding dimensionality; $e$ is the dimensionality of one-hot degree encoding for the degree features $\{\delta(v)\}$; $n_S := |\mathcal{W}^{(v)}|$ and $n_I := |\mathcal{W}_I^{(v)}|$ are the number of sampled RWs and number of RWs used to infer position embeddings for each node $v$; $\lambda_\psi$ and $\lambda_\gamma$ are learning rates to optimize identity and position embeddings; $m$ is the number of training iterations; in each iteration, we update identity and position embeddings $m_\psi$ and $m_\gamma$ times; $l$ is the RW length; $\alpha$ and $\tau$ are hyper-parameters in the training losses.

Tables 11 and 12 give layer configurations for the *transductive* and *inductive* embedding inference, where $\mathrm{Enc}_\varphi(\cdot)$ and $\mathrm{Dec}_\varphi(\cdot)$ denote the AW encoder and decoder described in (1); $\mathrm{Red}_s(\cdot)$ is the feature reduction

Table 13: Evaluation Results on Attributed Graphs for the Validation of Inconsistency of Attributes.

| | Cornell | | Texas | | Washington | | Wisconsin | |
|---|---|---|---|---|---|---|---|---|
| | Mod↑(%) | NCut↓ | Mod↑(%) | NCut↓ | Mod↑(%) | NCut↓ | Mod↑(%) | NCut↓ |
| node2vec | **56.93** | 3.18 | **45.99** | 3.10 | **44.94** | 3.59 | **54.73** | 2.97 |
| struc2vec | -9.50 | **1.53** | -11.37 | **1.17** | -9.33 | **0.71** | -8.94 | **1.34** |
| att-emb | -0.09 | 3.76 | -0.01 | 3.75 | -2.30 | 3.84 | -3.54 | 3.75 |
| [n2v∥att] | 50.80 | 3.38 | 35.26 | 3.12 | 44.05 | 3.50 | 54.02 | 3.13 |
| [s2v∥att] | -5.76 | 1.81 | -12.77 | 1.33 | -2.81 | 1.00 | -9.58 | 1.35 |

unit (2); $\mathrm{Dec}_\psi(\cdot)$ represents the identity embedding decoder (4); $\mathrm{ReAtt}(\cdot)$ is the attentive reweighting function (6); $\tilde{\eta}_l := |\tilde{\Omega}_l|$ is the reduced number of AWs; $h_\psi$, $h_{\mathrm{tran}}$, and $h_{\mathrm{rout}}$ represent the numbers of attention heads in identity embedding encoder (3), transformer encoder (8), and attentive readout function (9); $L_{\mathrm{tran}}$ is the number of transformer encoder layers; 't', 's', and 'r' denote the activation functions of Tanh, Sigmoid, and ReLU. For our IRWE method, we recommend setting $l \in \{4, 5, \cdots, 9\}$, $\alpha \in \{0.1, 0.5, 1, 5, 10\}$, $\tau \in \{1, 5, 10, 50, 100, 500, 1000\}$, and $m_\psi, m_\gamma \in \{1, 5, 10, 20\}$.

We adopted the standard multi-head attention (Vaswani et al., 2017) to build the identity embedding encoder (3) and attentive readout unit (9) of IRWE. An attention unit includes the inputs of key, query, and value described by $\mathbf{K} \in \mathbb{R}^{m \times d}$, $\mathbf{Q} \in \mathbb{R}^{n \times d}$, and $\mathbf{V} \in \mathbb{R}^{m \times d}$. Assume that there are $h$ attention heads. Let $\tilde{d} = d/h$. For the $j$-th head, we first derive linear mappings $\tilde{\mathbf{K}}^{(j)} = \mathbf{K}\mathbf{W}_k^{(j)}$, $\tilde{\mathbf{Q}}^{(j)} = \mathbf{Q}\mathbf{W}_q^{(j)}$, and $\tilde{\mathbf{V}}^{(j)} = \mathbf{V}\mathbf{W}_v^{(j)}$, with $\{\mathbf{W}_k^{(j)} \in \mathbb{R}^{d \times \tilde{d}}, \mathbf{W}_q^{(j)} \in \mathbb{R}^{d \times \tilde{d}}, \mathbf{W}_v^{(j)} \in \mathbb{R}^{d \times \tilde{d}}\}$ as trainable parameters. The attention head is defined as

$$\mathbf{Z}^{(j)} = \mathrm{Att}_j(\mathbf{Q}, \mathbf{K}, \mathbf{V}) := \mathrm{softmax}(\tilde{\mathbf{Q}}^{(j)}\tilde{\mathbf{K}}^{(j)T}/\sqrt{\tilde{d}})\tilde{\mathbf{V}}^{(j)}. \tag{21}$$

We further concatenate the outputs of all the heads via $\mathbf{Z} = \mathrm{Att}(\mathbf{Q}, \mathbf{K}, \mathbf{V}) := [\mathbf{Z}^{(1)} \| \cdots \| \mathbf{Z}^{(h)}]$.

All the experiments were conducted on a server with AMD EPYC 7742 64-Core CPU, 512GB main memory, and one NVIDIA A100 GPU (80GB memory). We used the official code or public implementations of all the baselines and tuned parameters to report their best performance. On each dataset, we set the same embedding dimensionality for all the methods.

# E  Further Experiment Results

To demonstrate the possible inconsistency of graph attributes for identity and position embedding as discussed in Section 1, we conducted additional experiments on four attributed graphs (i.e., *Cornell*, *Texas*, *Washington*, and *Wisconsin*) from WebKB[1]. For each graph, we extracted the largest connected component from its topology. After the pre-processing, we have $(N, E, M, K) = (183, 227, 1703, 5)$, $(183, 279, 1703, 5)$, $(215, 365, 1703, 5)$, and $(251, 450, 1703, 5)$ for *Cornell*, *Texas*, *Washington*, and *Wisconsin*, where $N$, $E$, and $K$ are numbers of nodes, edges, and clusters; $M$ is the dimensionality of node attributes.

We then applied node2vec and struc2vec, which are typical position and identity embedding baselines as described in Table 3, to the extracted topology of each graph, where we set embedding dimensionality $d = 64$. Furthermore, we derived special attribute embeddings (denoted as att-emb) with the same dimensionality by applying SVD to node attributes. Namely, we have three baseline methods (e.g., node2vec, struc2vec, and att-emb). To simulate the incorporation of attributes, we concatenated att-emb with node2vec and struc2vec, forming another two additional baselines denoted as [n2v∥att] and [s2v∥att]. The unsupervised community detection and node identity clustering were adopted as the downstream tasks for position and identity embedding, respectively. The evaluation results are depicted in Table 13, where att-emb outperforms neither (i) node2vec for community detection nor (ii) struc2vec for node identity clustering; the concatenation of att-emb cannot further improve the embedding quality of node2vec and struc2vec. The results imply that (i) *attributes may fail to capture both node positions and identities*; (ii) *the simple integration of attributes may even damage the quality of position and identity embeddings*.

---

[1] https://www.cs.cmu.edu/afs/cs/project/theo-20/www/data/

## F   Discussions of Future Research Directions

Some possible future research directions of this study are summarized as follows.

In this study, we focused on network embedding where topology is the only available information source without any attributes, due to the complicated correlations between the two sources (Qin et al., 2018; Li et al., 2019; Wang et al., 2020; Qin & Lei, 2021). We intend to explore the *adaptive incorporation of attributes*. Concretely, when attributes carry characteristics consistent with topology, one can fully utilize attribute information to enhance the embedding quality. In contrast, when there is inconsistent noise in attributes, we need to adaptively control the effect of attributes to avoid unexpected quality degradation.

In addition to mapping each node to a low-dimensional vector (a.k.a. node-level embedding), network embedding also includes the representation of a graph (a.k.a. graph-level embedding). We plan to extend IRWE to the graph-level embedding and evaluate the embedding quality for some graph-level downstream tasks (e.g., graph classification). To analyze the relations of graph-level embeddings to identity and position embeddings is also our next focus.

The optimization of IRWE adopts the standard full-batch setting, where we derive statistics or embeddings w.r.t. all the nodes when computing the training losses. This setting may not be scalable to graphs with large numbers of nodes. Inspired by recent studies of scalable GNNs (Zhang et al., 2022; Liu et al., 2023), we intend to explore a scalable optimization strategy based on mini-batch settings.

