# OpenReview forum: "IRWE: Inductive Random Walk for Joint Inference of Identity and Position Network Embedding"
_TMLR — Accepted by TMLR_

### Review · Reviewer_gxMC · 2024-05-11

**Summary Of Contributions:**

The work proposes a new method of learning node representations from random walks. The results have shown significant performance improvement over previous methods in several node classification tasks.

**Audience:**

Yes

**Broader Impact Concerns:**

No concerns.

**Claims And Evidence:**

No

**Requested Changes:**

Please make the writing concise and highlight the main idea.

**Strengths And Weaknesses:**

Strength:

The shown performance has a significant improvement over previous methods.

Weakness:

1. The methodology part of the work is very hard. The system is super complex, making it very hard to verify the correctness. If the model is engineered toward just improving performance over benchmark datasets, I am not sure what new knowledge I can learn from this work. Moreover, the main claimed contribution is the joint learning of identity embedding and positional embedding, but this has been considered by previous work such as PhUSION. The submission may need to be more specific about the motivation, aka, which problem is addressed by the proposed work.

2. I am not sure whether the learned embeddings are inductive. When the position embedding is used, how does the model generalize to a new graph? In particular, how does the model align a node in the new graph to a corresponding node in the training graph such the two nodes have "similar" positions and get similar embeddings?

3. The submission is too long because a lot of descriptions are redundant. Here I just give a few examples:
1) the discussion of embeddings in Section 1 has a lot of overlap with the discussion in Section 2.
2)Table 1 is not needed as long as they are clearly defined in the text.
3) the first paragraph in Section 4 before 4.1 is only distracting. It should be integrated into the discussion of the method.
4) Attention should not be a definition in the work because it is irrelevant to the work's main focus.

Overall, I feel that the work should be compressed to 12 pages.

4. The "formal" definitions and hypotheses are not accurate. For example, in Definition 1, there are quite a few ambiguous terms such as "some key properties". In definition 2: "the structural role". In definition 3, "the high-order linkage". In my view, a definition should be mathematically accurate.

Furthermore, theorem 1 from (Micali & Zhu, 2016) is interpreted incorrectly in this submission: $l = 2(m+1)$ is the complexity used only one induction step in the original work, not the complexity for the entire recovery algorithm.

---

> ### Author Response · Authors · 2024-07-30
> **Response to the Comments of Reviewer gxMC (1/3)**
>
> **Overall Comment**: The work proposes a new method of learning node representations from random walks. The results have shown significant performance improvement over previous methods in several node classification tasks.
>
> **Strength**: The shown performance has a significant improvement over previous methods.
>
> **General Response**: We sincerely thank you for reviewing our manuscript and have tried our best to make a major revision according to your concerns and suggestions regarding the weaknesses of our manuscript. In the rest of this comment, we give responses to your review comments one by one in detail and highlight our corresponding revisions.
>
> ***
> **W1**: The methodology part of the work is very hard. The system is super complex, making it very hard to verify the correctness. If the model is engineered toward just improving performance over benchmark datasets, I am not sure what new knowledge I can learn from this work.
>
> Moreover, the main claimed contribution is the joint learning of identity embedding and positional embedding, but this has been considered by previous work such as PhUSION. The submission may need to be more specific about the motivation, aka, which problem is addressed by the proposed work.
>
> **Rsp**: Please note that we consider a very challenging problem, i.e., **the (i) *joint learning* and (ii) *inductive inference* of identity and position embeddings (iii) *without relying on the availability and aggregation of attributes***. To the best of our knowledge, **we are the first to propose a possible solution to this challenging problem**, using *a sophisticated combination of multiple attention units with different designs*. To our knowledge, the well-known *AlphaFold* [1] adopts a similar design,  which combines multiple attention units with different definitions, for protein structure prediction.
>
> As summarized in **Section 2**, most existing embedding methods can *only support either identity embedding or position embedding*. Although some recent studies consider the relation between the two types of embeddings, they only focus on the learning of one type of embedding and the simple transform to another type, which *CANNOT support the joint learning of the two types of embeddings*.
>
> As we can check in the paper and official code of *PhUSION*, **it *CANNOT* support the *joint learning* and *inference* of identity and position embeddings**. Concretely, *PhUSION* is a framework including three steps (i.e., the computation of node proximities, non-linear filtering, and dimension reduction). Each unique setting of these steps **can only derive *either* identity *or* position embeddings** but ***CANNOT* simultaneously derive two sets of embeddings**. In this sense, *PhN-PPMI* and *PhN-HK* in our experiments are entirely different methods but not embeddings given by a method.
>
> [1] Jumper, John, et al. Highly Accurate Protein Structure Prediction with AlphaFold. Nature 2021.

---

> ### Author Response · Authors · 2024-07-30
> **Response to the Comments of Reviewer gxMC (2/3)**
>
> **W2**: I am not sure whether the learned embeddings are inductive. When the position embedding is used, how does the model generalize to a new graph? In particular, how does the model align a node in the new graph to a corresponding node in the training graph such the two nodes have "similar" positions and get similar embeddings?
>
> **Rsp**: To the best of our knowledge, **inductive graph inference is defined based on the *transferability of model parameters***. In particular, inductive inference should allow to **directly transfer/generalize model parameters**, which were trained on existing known graph topology, **to new unseen nodes or even across graphs**, **without any re-training**.
>
> When applying IRWE to a new topology, we (i) **sample random walks (RWs) starting from each new node**, (ii) **map these RWs to corresponding anonymous walks (AWs)**, and (iii) **(incrementally) derive induced statistics**. Then, we can **directly obtain the identity and position embeddings w.r.t. new nodes via *only one feed-forward propagation of IRWE***, with the sampled RWs, induced AWs, and statistics as inputs. Since we **do not need to re-train the IRWE model (i.e., updating its model parameters) in this inference procedure**, IRWE can support the inductive inference.
>
> Since nodes $(v, u)$ with similar positions are more likely to be reached via RWs starting from them, $(v, u)$ are expected to have similar induced RW statistics and thus similar (inductive) position embeddings. Please also note that **our evaluation of identity and position embeddings only involves nodes within the same graph** (e.g., a new unseen graph for inductive inference). In particular, **comparing the positions of nodes from different graphs is meaningless** because *node position is usually defined based on the distance or overlap of $l$-hop neighbors (in terms of community structures) between nodes*, which are meaningless for nodes from different graphs.
>
> We hope that the aforementioned explanations can answer your questions and help you better understand our method. To our knowledge, *CAW* [1] also adopts the temporal RWs and AWs to support the inductive temporal link prediction, following a motivation similar to our method.
>
> [1] Wang, Yanbang, et al. Inductive Representation Learning in Temporal Networks via Causal Anonymous Walks. ICLR 2021.

---

> ### Author Response · Authors · 2024-07-30
> **Response to the Comments of Reviewer gxMC (3/3)**
>
> **W3**: The submission is too long because a lot of descriptions are redundant. Here I just give a few examples:
> - The discussion of embeddings in Section 1 has a lot of overlap with the discussion in Section 2.
> - Table 1 is not needed as long as they are clearly defined in the text.
> - The first paragraph in Section 4 before 4.1 is only distracting. It should be integrated into the discussion of the method.
> - Attention should not be a definition in the work because it is irrelevant to the work's main focus.
>
> **Request Change**: Please make the writing concise and highlight the main idea.
>
> **Rsp**: We are sorry for this presentation issue of our manuscript. During the revision, we have tried our best to shorten the length of the main paper, although **Reviewer D6v2** and **Reviewer a8jk** also suggested to add more discussions and experiments. Some related revisions are summarized as follows.
> - In **Section 2**, we removed duplicated content that overlaps with **Section 1**.
> - We removed **Table 1 in our previous version** and ensured that all the notations were clearly defined in the text.
> - We removed some duplicated descriptions **at the beginning of Section 4 (in our previous version)** that will be detailed later in the rest of **Section 4**.
> - We moved preliminaries of multi-head attention from **Section 3 (of our previous version)** to **Appendix D (of the revised manuscript)**.
> - We moved the complexity analysis from **Section 4.4 (of our previous version)** to **Appendix B (of the revised manuscript)**.
> - We moved definitions of the NCut and modularity metrics from **Section 5.1 (of our previous version)** to **Appendix D (of the revised manuscript)**.
> - We moved descriptions of the experiment environment from **Section 5.1 (of our previous version)** to **Appendix D (of the revised manuscript)**.
> - We moved experiments regarding the inconsistency between graph topology and attributes from **Section 5.6 (of our previous version)** to **Appendix E (of the revised manuscript)**.
> - We moved discussions of future research directions from **Section 6 (of our previous version)** to **Appendix F (of the revised manuscript)**.
> - We redrew most of the figures (c.f. **Fig. 1, 2, and 3 in the revised manuscript**) to fully utilize the space of the TMLR template.
> - We reorganized the layout of some figures and tables (c.f. **Fig. 4, Table 1, Table 2, and Table 3 in the revised manuscript**) to fully utilize the space of the TMLR template.
>
> As a result, **the length of the main paper has been reduced by about 30%**. We really hope that our revisions can make the manuscript more concise and readable.
>
> ***
> **W4**: The "formal" definitions and hypotheses are not accurate. For example, in Definition 1, there are quite a few ambiguous terms such as "some key properties". In definition 2: "the structural role". In definition 3, "the high-order linkage". In my view, a definition should be mathematically accurate.
>
> **Rsp**: We are sorry for the presentation issue regarding formal definitions. During the revision, we have adopted your suggestion to reformulate some definitions.
>
> Concretely, we first define (i) **node identity** (a.k.a. structural role) and (ii) **node position** (a.k.a. linkage similarity) based on the (i) **rooted subgraph** $\mathcal{G}_s (v, l)$ of each node $v$ and (ii) **overlap of $l$-hop neighbors** between nodes in **Definition 1** and **Definition 2**, respectively. In **Definition 3**, we further highlight that we consider **unsupervised network embedding that preserves *either* node identities *or* node positions**, which are defined as **identity embedding** or **position embedding**, respectively.
>
> The corresponding revisions are highlighted with **blue text** and can be checked in **Section 3**. We hope that our revisions can help you better understand some key concepts (e.g., node identity, node position, identity embedding, and position embedding) of this study.
>
> ***
> **W5**: theorem 1 from (Micali & Zhu, 2016) is interpreted incorrectly in this submission: $l = 2(m+1)$ is the complexity used only one induction step in the original work, not the complexity for the entire recovery algorithm.
>
> **Rsp**: We are sorry for the incorrect interpretation of **Theorem 1**. After carefully checking the paper [1], we changed the statement 'one can reconstruct $\mathcal{G}_s (v, r)$ using $(q(v,1), \cdots,q(v, l))$, where l = 2(m+1)' in **Theorem 1** to 'one can reconstruct $\mathcal{G}_s(v, r)$ in time $O (n^2)$ with $O (n^2)$ access to $[q(v, 1), \cdots, q(v, l)]$, where $l = O(m)$; $n$ and $m$ are the numbers of nodes and edges in $\mathcal{G}_s(v, r)$'. The corresponding revisions are highlighted with **blue text**.
>
> [1] Micali, Silvio, and Zeyuan Allen Zhu. Reconstructing Markov Processes from Independent and Anonymous Experiments. Discrete Applied Mathematics 2016.

---

### Review · Reviewer_D6v2 · 2024-06-10

**Summary Of Contributions:**

This paper introduces a new embedding method acting as a unified framework for the joint inductive inference of identity and position embeddings where node attributes are not present. The paper introduces an inductive random walk embedding (IRWE) method that utilizes multiple attention mechanisms to create a random walk on graph topology defining jointly-optimized identity and position embeddings. Results based on multiple networks for in an inductive, as well as, transductive setting in the tasks of node classification and node identity clustering, showcase the superiority of the method against the considered baselines.

**Audience:**

Yes

**Broader Impact Concerns:**

No concerns.

**Claims And Evidence:**

Yes

**Requested Changes:**

1) The paper should discuss connections to and compare its performance with classical studies, such as the work by Peter D. Hoff.

2) The paper should include link prediction experiments, as this is a prominent downstream task.

3) More recent transductive methods should also be considered, as most of the methods currently considered are not very recent.

4) The paper should include a discussion on the number of parameters used by the proposed method and compare this with the considered baselines.

**Strengths And Weaknesses:**

Strengths:

1) The paper proposes a novel and interesting approach for characterizing identity and position embeddings based on random walks over the graph.

2) The model performs well in both inductive and transductive settings.

3) The paper includes multiple experiments across various settings, often significantly outperforming the baselines.

Weaknesses:

1) The paper does not discuss existing literature that relates capturing identity and positional embeddings to modeling stochastic equivalence and homophily in networks, such as "Modeling homophily and stochastic equivalence in symmetric relational data" by Peter D. Hoff, 2008.

2) The paper lacks any link prediction experiments, which is one of the most prominent downstream tasks.

3) The analysis and description of the method are somewhat convoluted and can be difficult to follow at times.

---

> ### Author Response · Authors · 2024-07-30
> **Response to the Comments of Reviewer D6v2 (1/3)**
>
> **Overall Comment**: This paper introduces a new embedding method acting as a unified framework for the joint inductive inference of identity and position embeddings where node attributes are not present. The paper introduces an inductive random walk embedding (IRWE) method that utilizes multiple attention mechanisms to create a random walk on graph topology defining jointly-optimized identity and position embeddings. Results based on multiple networks for in an inductive, as well as, transductive setting in the tasks of node classification and node identity clustering, showcase the superiority of the method against the considered baselines.
>
> **Strengths**:
> - The paper proposes a novel and interesting approach for characterizing identity and position embeddings based on random walks over the graph.
> - The model performs well in both inductive and transductive settings.
> - The paper includes multiple experiments across various settings, often significantly outperforming the baselines.
>
> **General Response**: We sincerely thank you for summarizing some strengths of our manuscript. Besides, we have also tried our best to make a major revision according to your comments and suggestions regarding the weaknesses. In the rest of this comment, we give detailed responses to each of your review comments and highlight our corresponding revisions.
>
> ***
> **W1**: The paper does not discuss existing literature that relates capturing identity and positional embeddings to modeling stochastic equivalence and homophily in networks, such as "Modeling homophily and stochastic equivalence in symmetric relational data" by Peter D. Hoff, 2008
>
> **Requested Change 1**: The paper should discuss connections to and compare its performance with classical studies, such as the work by Peter D. Hoff.
>
> **Rsp**: Thank you for recommending a significant related work [1]. During the revision, we have adopted your suggestion to briefly introduce this work in **Section 2**. In particular, we highlighted the following differences.
> - The *eigen-model* proposed in [1] is **a conventional probabilistic model** (with model parameters estimated via the MCMC algorithm) while we consider **unsupervised network embedding** in this study, which may involve the optimization and inference of some sophisticated deep learning structures.
> - According to our understanding, the *eigen-model* proposed in [1] can capture **either node positions or identities** (also defined as homophily and stochastic equivalence in [1]) under a unique parameter setting. In contrast, we explore **a unified framework for the *joint learning* and *inductive inference* of both identity and position embeddings**.
>
> In addition, we have also introduced and compared a concurrent work [2] that considers the underlying relation between identity and position embedding in **Section 2**.
>
> All the aforementioned revisions are highlighted with **blue text** and can be checked in **Section 2**.
>
> [1] Hoff, Peter. Modeling Homophily and Stochastic Equivalence in Symmetric Relational Data. NIPS 2007.
>
> [2] Yan, Yuchen, et al. PaCEr: Network Embedding From Positional to Structural. Web Conference 2024.

---

> ### Author Response · Authors · 2024-07-30
> **Response to the Comments of Reviewer D6v2 (2/3)**
>
> **W2**: The paper lacks any link prediction experiments, which is one of the most prominent downstream tasks.
>
> **Requested Change 2**: The paper should include link prediction experiments, as this is a prominent downstream task.
>
> **Rsp**：Thank you for your suggestion about additional experiments of link prediction. Please note that we focus on the *joint inductive inference of identity and position embeddings* in this study. To the best of our knowledge, **the relation between link prediction and identity (or position) embedding learning is unclear**.
>
> For instance, the survey paper [1] highlights that *identity embedding can achieve better link prediction quality* while some related studies [2,3] *use link prediction to verify the effectiveness of position embedding*. According to our understanding, the prior work [4] you mentioned in **W1** used link prediction as the downstream task. On some datasets, *the latent class model (w.r.t. node position) achieves better link prediction quality than the latent distance model (w.r.t. node identities)*. In contrast, *the latent distance model can also perform better on the rest datasets*.
>
> In this sense, **the quality of link prediction cannot fully measure the ability to capture node identities (or positions)**. Since **Reviewer gxMC** suggested to further shorten our manuscript, we have to *give a higher priority to report evaluation results highly related to identity and position embedding* (i.e., node identity classification, node position classification, node identity clustering, and community detection) in the main paper.
>
> Although we only have two weeks to revise our manuscript and give responses to reviewers, we still tried our best to conduct additional link prediction experiments. Some preliminary results regarding the prediction quality (in terms of AUC) of *node2vec* and *struc2vec*, which are typical position and identity embedding methods, are shown as follows.
>
> |           | PPI   | Wiki  | BlogCatalog | USA   | Europe | Brazil | Cora  | Citeseer | Cornell | Texas  | Washington | Wisconsin |
> |-----------|-------|-------|-------------|-------|--------|--------|-------|----------|---------|--------|------------|-----------|
> | node2vec  | 79.52 | 88.67 | 87.48       | 82.03 | 82.89  | 82.73  | **92.42** | **99.97**    | **100.00**  | **100.00** | 99.34      | **100.00**    |
> | struc2vec | **84.16** | **89.35** | **89.40**       | **87.75** | **85.29**  | **83.99**  | 68.92 | **99.97**    | 98.95   | **100.00** | **100.00**     | **100.00**    |
>
> Although *struc2vec (w.r.t. identity embedding) can achieve better quality in most cases*, *node2vec (w.r.t. position embedding) can also perform better on some datasets*. In some cases, the two methods can achieve very close or even the same prediction quality. In summary, **our experiment results also validate the aforementioned discussions regarding the unclear relation between link prediction and identity (or position) embedding**.
>
> [1] Rossi, Ryan A., et al. On Proximity and Structural Role-based Embeddings in Networks: Misconceptions, Techniques, and Applications. TKDD 2020.
>
> [2] Grover, Aditya, and Jure Leskovec. node2vec: Scalable Feature Learning for Networks. KDD 2016.
>
> [3] Yan, Yuchen, et al. PaCEr: Network Embedding From Positional to Structural. Web Conference 2024.
>
> [4] Hoff, Peter. Modeling Homophily and Stochastic Equivalence in Symmetric Relational Data. NIPS 2007.
>
> ***
> **W3**: The analysis and description of the method are somewhat convoluted and can be difficult to follow at times.
>
> **Rsp**: We are sorry for this presentation issue of our manuscript. During the revision, we have comprehensively adopted the suggestions of all the reviewers to **simplify descriptions of the methodology** and **shorten the main paper**.
>
> As a result, **the length of the main paper has been reduced by about 30%**. Please refer to our response to **W3** of **Reviewer gxMC** to check detailed revisions. We really hope that our revisions can make the manuscript more concise and readable.

---

> ### Author Response · Authors · 2024-07-30
> **Response to the Comments of Reviewer D6v2 (3/3)**
>
> **Requested Change 3**: More recent transductive methods should also be considered, as most of the methods currently considered are not very recent.
>
> **Rsp**: Thank you for your suggestion regarding baseline methods. Please note that **all the *inductive* methods can support the corresponding *transductive* embedding inference**. **To obtain the (transductive) evaluation results reported in *Tables 4 and 5 (of the revised manuscript)*, all the methods were treated as transductive ones**. Our experiments have already included some recent (inductive) methods. As summarized in **Table 3 (of the revised manuscript)**, *GraphMAE* and *UGFormer* are methods proposed in 2022 while *GraphMAE2* and *CSGCL* were published in 2023.
>
> During the revision, we have also adopted your suggestion to **include a recent *transductive* method *PaCEr* [1] published in 2024**. *PaCEr* is a concurrent work that also considers the relation between identity and position embeddings. It first optimizes position embeddings based on the observed graph topology and simply transforms them into identity embeddings. In contrast, our IRWE method considers the joint learning and inductive inference of both identity and position embeddings. The corresponding results of *PaCEr* can be checked in **Tables 4 and 5 (of the revised manuscript)**.
>
> [1] Yan, Yuchen, et al. PaCEr: Network Embedding From Positional to Structural. Web Conference 2024.
>
> ***
> **Requested Change 4**: The paper should include a discussion on the number of parameters used by the proposed method and compare this with the considered baselines.
>
> **Rsp**: Thank you for your suggestion about the complexity of parameters. During the revision, we summarized and compared the complexities of model parameters to be learned for all the methods in **Table 8 (of the revised manuscript)** and gave corresponding discussions in **Appendix B (of the revised manuscript)**. The corresponding revisions are highlighted with **blue text**.

---

### Review · Reviewer_a8jk · 2024-07-16

**Summary Of Contributions:**

This paper introduces a new model for unsupervised graph embedding. The model extracts both an identity embedding and a position embedding for each node of a graph. The method is inductive (i.e. a trained model can be used to extract embeddings for a new unseen graph).
The method relies on random walks and the related concept of anonymized walks.

**Audience:**

Yes

**Claims And Evidence:**

No

**Requested Changes:**

Related to the weaknesses pointed above, I am requesting two major changes:

1. If my understanding of your evaluation (cf above) is correct, it is required that you either:
    * report IRWE performance for a fixed set of hyper-parameters
    * cross-validate hyper-parameters on the validation set for both IRWE and the baselines and report resulting performance
2. Your illustrations should be revamped to better help the reader in their understanding of your notations

**Strengths And Weaknesses:**

* Strengths
    * [S1] The method is clearly positioned wrt the state of the art
    * [S2] The authors take care to motivate their technical decisions (using Theorem 1 and Hypotheses 1 and 2, among others), even if some decisions could be even further explained (cf below)
* Weaknesses
    * [W1] I have a major problem with the evaluation as it is now: it seems performance reported is based on hyper-parameter tuning, but there is no discussion in the text about how to select those hyper-parameters
    * [W2] The overall architecture is rather complex. Though the authors try to break the whole architecture in smaller bits to better explain them (which is really appreciated), it is sometimes hard to follow and some of the illustrations can be misleading

More detailed comments regarding the above-mentioned weaknesses:

* W1
    * In cases where supervised information is available, it would be nice to provide performance for some supervised methods (GAT & co): even if one does not expect your unsupervised approach to compete with those, it is always interesting to have an idea of how large the gap is on downstream tasks
    * You should use bold + underline for all methods in your tables for the reader to make her opinion
    * It would be interesting to report performance for the IRWE method in case only one of the embeddings is used (position or identity) in all tables 5 to 7
    * Performance reported in Tables 5 to 7 relies on hyper-parameter selection that seems to have been performed based on test set performance. Tables 10 to 13 report the hyper-parameters used, yet no hyper-parameter selection scheme is presented. If you compare your method to a bunch of baselines, you need to either use a fixed set of hyper-parameters that works well across a large number of datasets or provide a hyper-parameter selection method (e.g. rely on the validation set for selection)
* W2
    * In Fig. 3: the "one-hot encoding" is a bit strange: it seems the value 1 is sometimes encoded as 0100 and sometimes as 0001, depending on the index at which it occurs (eg compare [2] and [6]). Another reason why your one-hot encoding is not very intuitive is that the value 0 is encoded as "0000" whereas 1 is encoded as "0100": it seems the first bit is never used
    * In practice, it seems you do not have $d << l^2$, hence Fig.4 is misleading as the AW Encoder is not really contracting information
    * Fig. 4 is incomplete since the identity Embedding Regularization unit reconstructs $s(v)$ too, I think. Also, it is unclear why this is called a regularization unit and not a decoder, as for the AW decoder...
    * Some points on which the paper does not provide sufficient intuition
        * Why would information from AW statistics and degree information be complementary? The paper lacks intuition about many such questions that should be answered to motivate the chosen architecture
        * Wouldn't it be better to resample RWs every few epochs, in order to better catch information from a node's neighbourhood?

---

> ### Author Response · Authors · 2024-07-30
> **Response to the Comments of Reviewer a8jk (1/3)**
>
> **Overall Comment**: This paper introduces a new model for unsupervised graph embedding. The model extracts both an identity embedding and a position embedding for each node of a graph. The method is inductive (i.e. a trained model can be used to extract embeddings for a new unseen graph). The method relies on random walks and the related concept of anonymized walks.
>
> **Strengths**
>
> [S1] The method is clearly positioned wrt the state of the art
>
> [S2] The authors take care to motivate their technical decisions (using Theorem 1 and Hypotheses 1 and 2, among others), even if some decisions could be even further explained (cf below)
>
> **General Response**: We sincerely thank you for summarizing the strengths of our manuscript. In addition, we have also tried our best to make a major revision according to your comments and suggestions regarding the weaknesses. In the rest of this comment, we give detailed responses to your review comments one by one and highlight our corresponding revisions.
>
> ***
> **W1.1**: In cases where supervised information is available, it would be nice to provide performance for some supervised methods (GAT & co): even if one does not expect your unsupervised approach to compete with those, it is always interesting to have an idea of how large the gap is on downstream tasks
>
> **Rsp**: Thank you for your comment about the consideration of supervised baselines. Please note that **we consider the unsupervised network embedding** in this study as stated in Section 3. From our perspective, it is unfair to compare an unsupervised method with supervised baselines, which incorporate additional supervised label information for embedding learning. Since **Reviewer gxMC** also suggested us to further shorten our manuscript, we have to give a higher priority to report the evaluation results of unsupervised methods in the main paper.
>
> During the revision, we also considered your suggestion to include some supervised baselines in the appendix. However, after carefully checking of the official code of some well-known supervised baselines (e.g., GCN and GAT), we found that **their evaluation pipeline *could not* be directly applied to our experiments**. Please note that some datasets (e.g., PPI, Wiki, and BlogCatalog) in our experiments are with ground-truth of *multi-label node classification*, where *each node can simultaneously belong to more than one class*. In contrast, the standard evaluation pipeline of most supervised baselines *only allows each node to belong to one unique class*. Moreover, most supervised baselines also adopt a data split strategy different from ours. We really hope that you can understand the aforementioned difficulties we encountered.
>
> ***
> **W1.2**: You should use bold + underline for all methods in your tables for the reader to make her opinion
>
> **Rsp**: Thank you for your comment regarding the presentation of the experiment results. During the revision, we have adopted your suggestion to **use the 'bold + underline' presentation for all the methods**, where metrics are in bold or underlined if they perform the best or within top-3. The corresponding revisions can be checked in **Tables 4-6** of the revised manuscript.
>
> ***
> **W1.3**: It would be interesting to report performance for the IRWE method in case only one of the embeddings is used (position or identity) in all tables 5 to 7
>
> **Rsp**: Thank you for your comment regarding the additional evaluation results of IRWE. During the revision, we have adopted your suggestion to **report the results of IRWE's identity embedding and position embedding** (denoted as **IRWE**($\psi$) and **IRWE**($\gamma$)) **w.r.t. all the tasks**. The corresponding revisions are highlighted with **blue text** in **Tables 4-6** of the revised manuscript.
>
> In particular, **IRWE**($\psi$) and **IRWE**($\gamma$) have poor performance on tasks regarding node positions and identities, respectively. It further validates our motivation that *identity and position embeddings capture two different properties that may contradict with one another*.

---

> ### Author Response · Authors · 2024-07-30
> **Response to the Comments of Reviewer a8jk (2/3)**
>
> **W1.4**: Performance reported in Tables 5 to 7 relies on hyper-parameter selection that seems to have been performed based on test set performance. Tables 10 to 13 report the hyper-parameters used, yet no hyper-parameter selection scheme is presented. If you compare your method to a bunch of baselines, you need to either use a fixed set of hyper-parameters that works well across a large number of datasets or provide a hyper-parameter selection method (e.g. rely on the validation set for selection)
>
> **Requested Change 1**: If my understanding of your evaluation (cf above) is correct, it is required that you either:
> - report IRWE performance for a fixed set of hyper-parameters
> - cross-validate hyper-parameters on the validation set for both IRWE and the baselines and report resulting performance
>
> **Rsp**: Thank you for your comment and suggestion regarding the setting of hyper-parameters. Please note that *our data split includes a validation set with 10% nodes* (if ground-truth is available) and *we repeated the data split procedure 10 times with the average metrics reported*, as stated in **Sections 5.2** and **5.3**. After carefully checking our code, we found that **our evaluation has already followed a strategy similar to the 10-fold cross-validation**, where we (i) *randomly split the node set into $10$ subsets with each one used as the validation set in a round* and (ii) *used the average quality w.r.t. the validation set to tune parameters of all the methods*.
>
> During our revision, we have also added some more descriptions regarding the data splitting and parameter settings in **Sections 5.2** and **5.3**, with the corresponding revisions highlighted with **blue text**.
>
> ***
> **W2.1**: In Fig. 3: the "one-hot encoding" is a bit strange: it seems the value 1 is sometimes encoded as 0100 and sometimes as 0001, depending on the index at which it occurs (eg compare [2] and [6]). Another reason why your one-hot encoding is not very intuitive is that the value 0 is encoded as "0000" whereas 1 is encoded as "0100": it seems the first bit is never used
>
> **Rsp**: We are sorry that there is a typo for the one-hot encoding of AWs in **Fig. 3**. During the revision, we have corrected this typo, where **value 1 must be encoded as '0100'**.
>
> We also noticed that the first bit of one-hot encoding is always 0. We still adopted this design because **one-hot encoding is a straightforward representation of sequential data that can be handled by typical deep neural network structures**. Our experiments demonstrate that IRWE can still effectively support the joint inductive inference of identity and position embeddings, even with this naive design that includes some redundant bits.
>
> By considering the special property of a feasible AW $\omega = (I_{\omega}(w^{(0)}), \cdots, I_{\omega}(w^{(l)}))$, where we must have $I_{\omega}(w^{(i)}) \le I_{\omega}(w^{(i+j)})$, there exist more compressed representations. As stated in **Appendix F** (of the revised manuscript), to improve the scalability of IRWE is a significant future direction of this study. We also intend to explore a more compressed AW encoding in our future work, which can help improve the scalability.
>
> ***
> **W2.2**: In practice, it seems you do not have $d \ll l^2$, hence Fig.4 is misleading as the AW Encoder is not really contracting information
>
> **Rsp**: Thank you for your comment regarding the model design and presentation of **Fig. 4 (in our previous version)**. You are correct. We cannot ensure that $d \ll l^2$. During the revision, we have changed the presentation of the AW encoder, which does not contract information. The corresponding revisions can be checked in **Fig. 2 (b) of the revised manuscript**.
>
> ***
> **W2.3**: Fig. 4 is incomplete since the identity Embedding Regularization unit reconstructs $s(v)$ too, I think. Also, it is unclear why this is called a regularization unit and not a decoder, as for the AW decoder...
>
> **Rsp**: We are sorry for the presentation issue of **Fig. 4 (in our previous version)**. During the revision, we have adopted your suggestion to redraw the architecture of the identity embedding module, where both AW statistics $\{ s(v) \}$ and high-order degree features $\{ \delta (v) \}$ are inputs of the feature reduction unit and are further reconstructed by this module.
>
> To achieve a clearer presentation, we **have also changed the terminologies** (i) *AW embedding derivation*, (ii) *identity embedding derivation*, (iii) *identity embedding regularization*, (iv) *position embedding derivation*, and (v) *position embedding regularization* **to** (i) *AW embedding auto-encoder*, (ii) *identity embedding encoder*, (iii) *identity embedding decoder*, (iv) *position embedding encoder*, and (v) *position embedding decoder*, respectively.
>
> The corresponding revisions can be checked in **Fig. 2** of the revised manuscript.

---

> ### Author Response · Authors · 2024-07-30
> **Response to the Comments of Reviewer a8jk (3/3)**
>
> **W2.4**: Some points on which the paper does not provide sufficient intuition:
> - Why would information from AW statistics and degree information be complementary?
> - Wouldn't it be better to resample RWs every few epochs, in order to better catch information from a node's neighborhood?
>
> **Rsp**: Please note that **AW statistics and high-order degree features are based on *two perspectives* of defining and characterizing node identities**, which are described in **Theorem 1** and **Hypothesis 1**, respectively. On the one hand, we can characterize the identity of each node using its *rooted subgraph*, which can be further characterized by the *induced AW statistics* based on **Theorem 1**. On the other hand, **Hypothesis 1** is also commonly adopted by prior work [1,2], which characterizes node identities using *degree statistics w.r.t. high-order neighbors*. To the best of our knowledge, there are no related studies that explicitly reveal the relation between these two perspectives.
>
> Since **IRWE can only *estimate* related statistics but cannot obtain their exact values**, we believe that **the combination of these two types of statistics can help resist the inconsistency of estimation**. Our ablation studies in **Section 5.4** also demonstrate that
> - (i) the case only considering one type of statistics can achieve quality close to IRWE for tasks regarding identity embedding;
> - (ii) the combination of these two types of statistics can help further improve the embedding quality.
>
> Compared with the feed-forward propagation and back-propagation, which can be significantly speeded up via GPUs, **the RW sampling is a major bottleneck for the training and inference efficiency of IRWE**, even though we have already adopted the parallel implementation of RW sampling. Although resampling RWs may potentially improve the embedding quality (as suggested in your comments), we still use a fixed set of RWs due to **efficiency concerns**. Our experiments also demonstrate that even without resampling, IRWE can achieve superior embedding quality over various baselines. A possible solution to this (efficiency) issue is to apply some **approximated RW sampling strategies** (e.g., the *path sampling* in [3]). We intend to consider this extension in our future work.
>
> During the revision, we have also added some more intuitive explanations for the aforementioned designs. We hope that our explanations can help you better understand our method.
>
> [1] Ribeiro, Leonardo FR, et al. struc2vec: Learning Node Representations from Structural Identity. KDD 2017.
>
> [2] Wu, Jun, et al. DEMO-Net: Degree-specific Graph Neural Networks for Node and Graph Classification. KDD 2019.
>
> [3] Qiu, Jiezhong, et al. NetSMF: Large-Scale Network Embedding as Sparse Matrix Factorization. WWW 2019.
>
> ***
> **Requested Change 2**: Your illustrations should be revamped to better help the reader in their understanding of your notations.
>
> **Rsp**: Thank you for your suggestion regarding the presentation of our manuscript. During the revision, we have made a series of revisions for your comments **W2.1**-**W2.3** (cf. our aforementioned responses to check the corresponding revisions).
>
> In addition, we have also comprehensively adopted the suggestions of all the reviewers to simplify descriptions of the methodology and shorten the main paper. As a result, the length of the main paper has been reduced by about 30%. Please refer to our response to **W3** of **Reviewer gxMC** to check detailed revisions.

---

### Author Response · Authors · 2024-07-30
**Summary of Major Revisions**

Dear Editor and Reviewers:

We appreciate your kind consideration of our manuscript TMLR#2393, entitled "IRWE: Inductive Random Walk for Joint Inference of Identity and Position Network Embedding". We are also grateful for the anonymous reviewers' insightful comments and constructive suggestions that helped us achieve a better presentation of our manuscript. We have comprehensively adopted the suggestions of all the reviewers and tried our best to make a major revision according to their concerns.

All the major revisions have been highlighted with **blue text** in the newly submitted version of our manuscript. For convenience, we summarize our major revisions as follows.
- In **Section 2**, we removed duplicated content that overlaps with **Section 1**.
- We introduced some more classic and recent studies regarding the relation between identity and position embedding in **Section 2.1**.
- We removed **Table 1 in our previous version** and ensured that all the notations were clearly defined in the text.
- We reformulated the definitions of (i) node identity, (ii) node position, and (iii) network embedding in **Definition 1, Definition 2, and Definition 3**.
- We moved preliminaries of multi-head attention from **Section 3 (of our previous version)** to **Appendix D (of the revised manuscript)**.
- We changed the the terminologies (i) *AW embedding derivation*, (ii) *identity embedding derivation*, (iii) *identity embedding regularization*, (iv) *position embedding derivation*, and (v) *position embedding regularization* **to** (i) *AW embedding auto-encoder*, (ii) *identity embedding encoder*, (iii) *identity embedding decoder*, (iv) *position embedding encoder*, and (v) *position embedding decoder*, respectively. The corresponding revisions can be checked in **Fig. 2 (of the revised manuscript)** and subtitles of **Section 4**.
- We removed some duplicated descriptions **at the beginning of Section 4** that will be detailed later in the rest of **Section 4**.
- We corrected the interpretation of **Theorem 1**.
- We moved the complexity analysis from **Section 4.4 (of our previous version)** to **Appendix B (of the revised manuscript)**.
- We moved definitions of the NCut and modularity metrics from **Section 5.1 (of our previous version)** to **Appendix D (of the revised manuscript)**.
- We moved descriptions of the experiment environment from **Section 5.1 (of our previous version)** to **Appendix D (of the revised manuscript)**.
- We added a new transductive embedding baseline (i.e., *PaCEr*) in **Section 5.1 and Section 5.2 (of the revised manuscript)**.
- We used the 'bold + underlined' presentation for all the methods in **Tables 4, 5, and 6 (of the revised manuscript)**.
- We reported results of the identity and position embeddings given by IRWE (denoted as **IRWE**($\psi$) and **IRWE**($\gamma$)) for all the tasks in **Tables 4, 5, and 6 (of the revised manuscript)**.
- We added some more descriptions regarding the data splitting and parameter tunning in **Section 5.2 and Section 5.3 (of the revised manuscript)**.
- We moved experiments regarding the inconsistency between graph topology and attributes from **Section 5.6 (of our previous version)** to **Appendix E (of the revised manuscript)**.
- We moved discussions of future research directions from **Section 6 (of our previous version)** to **Appendix F (of the revised manuscript)**.
- We summarized and compared the complexities of model parameters to be learned for all the methods in **Table 8 (of the revised manuscript)** and gave corresponding discussions in **Appendix C (of the revised manuscript)**.
- We redrew most of the figures (c.f. **Fig. 1, 2, and 3 in the revised manuscript**) to fully utilize the space of the TMLR template.
- We reorganized the layout of some figures and tables (c.f. **Fig. 4, Table 1, Table 2, and Table 3 in the revised manuscript**) to fully utilize the space of the TMLR template.

In addition, we also give responses to all the anonymous reviewers.

---

### Decision · Action_Editor_YtZe · 2024-09-18

**Recommendation:** Accept as is

**Comment:**

This work presents a method for unsupervised graph embedding without attributes. The proposed method jointly learns positional and identity embeddings based on an inductive random walk embedding method.

The approach is novel and seems to perform well in practice, and the evaluation is thorough. The reviewers had a number of concerns about the experimental settings, the exposition, and the discussion of related work, but these have all been adequately addressed in the revision. We appreciate the thorough discussion of the points raised.

While we appreciate that you have addressed concerns about the paper length, we suggest returning the figures to their original size, rather than shrinking them.

**Audience:**

Yes, this is of interest to many of TMLR's audience.

**Claims And Evidence:**

The proposed approach is well explained and thoroughly evaluated. The authors show improved performance against a number of strong baselines, and carry out a thorough ablation. The methodology is well explained and justified.